



# Black carbon aerosols in China: Spatial-temporal variations and lessons from long-term atmospheric observations

Huang Zheng[1], Shaofei Kong[2], Deping Ding[3, 4], Marjan Savadkoohi[5, 6], Congbo Song[7], Mingming Zheng[8], and Roy M. Harrison[9, 10]

[1]College of Resource and Environmental Engineering, Wuhan University of Science and Technology, 430081 Wuhan, China
[2]Department of Atmospheric Sciences, School of Environmental Studies, China University of Geosciences, Wuhan, 430078, China
[3]Beijing Weather Modification Center, Beijing, 100089, China
[4]Beijing Key Laboratory of Cloud, Precipitation and Atmospheric Water Resources, Beijing, 100089, China
[5]Institute of Environmental Assessment and Water Research (IDAEA-CSIC), 08034, Barcelona, Spain
[6]Department of Mining, Industrial and ICT Engineering (EMIT), Manresa School of Engineering (EPSEM), Universitat Politécnica de Catalunya (UPC), Manresa, 08242, Spain
[7]National Centre for Atmospheric Science (NCAS), Department of Earth and Environmental Science, The University of Manchester, Manchester M13 9PL, U.K.
[8]School of Chemical and Environmental Engineering, Wuhan Polytechnic University, Wuhan, 430023, China
[9]School of Geography, Earth and Environment Sciences, University of Birmingham, Birmingham B15 2TT, U.K.
[10]Department of Environmental Sciences, Faculty of Meteorology, Environment and Arid Land Agriculture, King Abdulaziz University, PO Box 80203, Jeddah, Saudi Arabia

**Correspondence:** Shaofei Kong (kongshaofei@cug.edu.cn) and Deping Ding (zytddp@vip.sina.com)

**Abstract.** Using 13-year (2008–2020) continuous measurements of equivalent black carbon (eBC) in China, this study reports the spatial-temporal variations of eBC and its sources, including solid fuel ($eBC_{sf}$) and liquid fuel combustion ($eBC_{lf}$). The results show that eBC and its sources exhibit spatial heterogeneity with higher concentrations in eastern and northern China compared to western and southern China. Seasonal variations of eBC and $eBC_{sf}$ generally show low values during summer and high values during winter in most stations. Long-term trends indicate that eBC and $eBC_{lf}$ decreased most rapidly at urban stations while $eBC_{sf}$ declined faster at rural stations. Comparisons of eBC concentrations and trends between this study and other observations reveal higher eBC levels but lower reduction rates in China. Comparison between surface eBC observations and model simulations indicates models performed well in simulating spatial distribution but poorly in simulating inter-annual variations. Weather-normalized eBC concentrations were compared to several emission inventories, revealing higher correlations and suggesting that normalized eBC concentrations can be used to adjust emission estimates. Long-term observations of eBC and its sources show decreasing trends in China, primarily driven by emission reduction. Further analysis suggests that the reduction of eBC was mainly attributed to decreased emissions from solid fuel combustion in rural and baseline stations. This study provides insights for reducing uncertainties in black carbon emission inventories and improving model performance in simulating surface concentrations.





# 1 Introduction

Despite being a small fraction of the aerosol chemical composition (Bond et al., 2013; Huang et al., 2014; Tao et al., 2017), black carbon (BC) is important for climate, air pollution, and human health. Globally, BC can cause up to +0.41 W m$^{-2}$ effective radiative forcing (Szopa et al., 2021). At the regional scale, BC aerosol can suppress the deepening of the planetary boundary layer and therefore enhance haze pollution (Ding et al., 2016; Wang et al., 2018) as well as radiative forcing (Peng

et al., 2016). Personal exposure to BC can cause adverse health effects such as carcinogenic risks and elevated blood pressure, and BC can reach the fetal side of the human placenta (Mordukhovich et al., 2009; Bové et al., 2019; Lin et al., 2019). As a short-lived climate forcer, the atmospheric lifetime of BC is about 5.5 days $\pm$ 35% (median $\pm$ 1 standard deviation, hereafter) according to an ensemble of 14 models (Gliß et al., 2021). Therefore, the study of BC aerosol is important to reduce its adverse impacts.

To better understand the climate, environmental, and health effects of BC, modeling its atmospheric abundance is important. Although Chemistry Transport Models (CTMs) can capture spatial-temporal variations of BC, the simulation accuracy is often poor and there are discrepancies among different simulations. For instance, 17 global models overestimated the abundance of BC, with the ratio of simulated to observed values ranging from 0.5 to 10 in Europe (Koch et al., 2009). Similarly, Ikeda et al. (2022) evaluated the performance of six BC emission inventories and found an overestimation of modeled BC concentration

by a factor of 1.24 to 2.16 in China. On the contrary, simulations underestimated the annual mean BC levels in China, with normalized mean bias ranging from $-83\%$ to $-22\%$ (Mao et al., 2016). The high uncertainties in the simulation of BC arise from the emission inventory and physical mechanisms. One method to constrain the model results is to use long-term observation datasets to evaluate the emission inventory, transport, and removal processes (Bond et al., 2013; Wang et al., 2014; Alvarado et al., 2016; He et al., 2016; Evangeliou et al., 2018). For instance, Evangeliou et al. (2018) coupled BC observations with

the top-down method to constrain the emission, and the posterior simulated BC concentrations showed better agreement with observations. Choi et al. (2020b) utilized long-term observations of BC to CO ratios to validate the regional emission inventory in Asia. They found that emissions in East and South Korea were overestimated, while emissions in North Korea were underestimated. Using long-term observations of BC and CO from 2009 to 2015 at Fukue Island, Kanaya et al. (2016) constrained the wet removal rate of BC and proposed a stretched exponential decay equation to describe the wet removal rate. Based on BC

observations from 2010 to 2016 at three representative background sites in East Asia, Choi et al. (2020a) estimated the average transport efficiency of BC to be 0.73, which was lower than the mean rate of 0.91 from the FLEXible PARTicle Lagrangian transport model (version 10.4).

Long-term observation of BC aerosol also contributes to a better understanding of the effectiveness of air quality management (Chen et al., 2016; Font and Fuller, 2016; Fuller and Font, 2019). For instance, a decreasing trend of BC from 2010 to

2014 in London was attributed to the application of diesel particle filters (Font and Fuller, 2016). Long-term observation of BC at 16 sites from 2009 to 2018 in Germany showed the highest decreases at roadside and urban background sites, implying strong evidence of reduced traffic emissions in urban areas (Sun et al., 2020). The decreasing trend of BC at traffic site in the Helsinki metropolitan area suggested the success of vehicle exhaust particle mitigation (Luoma et al., 2021). In China, there



were similar conclusions. For instance, a 38% reduction in BC concentration was observed in Beijing between 2005 and 2013,

primarily due to the relocation of industries and a decrease in consumption of coal and coke (Chen et al., 2016). Continuous observations of BC in Beijing from 2012 to 2020 revealed a 71% reduction, with the largest decrease occurring at nighttime. This suggests that pollution control measures implemented since 2013 have effectively reduced primary emissions (Sun et al., 2022). The long-term trend of BC in Wuhan suggested the positive role of pollution control measures across China on BC reduction at the observational site (Zheng et al., 2020). Continuous observation of BC at Mt. Waliguan showed a decreasing

rate of 2% yr$^{-1}$ from 2008 to 2017, which reflected the emission reduction of BC in China (Dai et al., 2021).

As summarized above, long-term observation of BC is important to improve CTM simulations and evaluate the effectiveness of air pollution control measures on air pollutants. Long-term surface observations of BC aerosol have been widely reported (Hirdman et al., 2010; Boreddy et al., 2018; Kutzner et al., 2018). For instance, the earliest continuous in situ observation of BC was launched at 150 urban and rural stations in 1962 in the United Kingdom (Novakov and Hansen, 2004). Compared to

long-term observations at single or a small number of stations, a monitoring network consisting of many stations can provide more information about spatial-temporal variations of air pollutants. However, long-term in situ observation networks are few. At present, the world-wide monitoring networks include the World Meteorological Organization Global Atmosphere Watch (WMO-GAW) (Bond et al., 2013), European Monitoring and Evaluation Programme (EMEP) (Yttri et al., 2007), German Ultrafine Aerosol Network (GUAN) (Birmili et al., 2016), UK Black Carbon Network (Singh et al., 2018), National Air

Pollution Monitoring Network (NABEL) in Switzerland (Grange et al., 2020), Interagency Monitoring of Protected Visual Environments network (IMPROVE) in the US (Malm et al., 1994), Canadian Aerosol Baseline Measurement (CABM) (Chan et al., 2019), Aerosol Radiative Forcing over India Network (ARFINET) in India (Manoj et al., 2019), and China Black Carbon Observational Network (CBNET) (Zhang et al., 2019b). Research from these observational networks is mainly focused on the spatial-temporal variations of BC (Birmili et al., 2016; Manoj et al., 2019; Zhang et al., 2019b; Savadkoohi et al., 2023). More

valuable information from long-term observations, however, remains untapped.

Therefore, this study presented a comprehensive BC dataset collected from CBNET covering the period from 2008 to 2020. Spatial-temporal characteristics, sources, and long-term trends of BC are reported. The information behind the results is explored with a focus on three questions: (1) what can we learn from the comparison between BC ground observations and CTM simulations; (2) whether the inter-annual variations of BC can be used as an indicator of BC emissions; (3) which factors

dominated the variations of BC in China during the past 13 years. By digging into the data from continuous measurements of BC in China, this study is expected to provide insights into reducing the uncertainties in BC emissions and simulations from CTMs.

## 2   Methodology

### 2.1   Observation sites

The CBNET, established in 2006, is a part of the China Atmosphere Watch Network (CAWNET). There are 68 stations across China and the number of stations has increased recently. In this study, 48 sites with observations during 2008-2020 were





selected (**Fig. S1**). According to the land type and surrounding environment , these 48 sites can be divided into baseline (N = 7), urban (N = 23), and rural (N = 18) stations (Guo et al., 2020; Zhang et al., 2008, 2019b). The baseline stations are situated around 100 km distant from local sources or nearby major cities, and they are positioned at a moderate height above
the local elevation of the area. For the urban stations, the sampling heights are deliberately set to be 50–100 m higher than the average elevation of the city. This strategic placement aims to collect samples that represent the entire region rather than just the immediate local surroundings (Zhang et al., 2008). More details about the selected stations can be found in **Table S1**.

### 2.2 Instruments and data prepossessing

The instruments used in each monitoring site are provided in **Table S1**. The BC mass concentration was measured by the
filter-based absorption Aethalometers. Most of the stations utilized the AE31 to measure the BC through the entire period (i.e., 2008–2020) and some stations used both the AE31 and AE33 (**Table S1**). The details about the principles, operations, and algorithms of AE31 and AE33 can be found elsewhere (Drinovec et al., 2015; Rajesh and Ramachandran, 2018). Briefly, ambient aerosol is drawn through a cyclone with a cut-off size of 2.5 μm at a flow rate of 2–5 L min$^{-1}$ and deposited on a quartz fiber tape (AE31) or tetrafluoroethylene-coated glass filter (AE33). Changes in attenuation before and after sampling
are measured to be converted into aerosol absorption coefficients ($b_{abs}$) and then into mass concentrations using the pre-defined mass attenuation cross-section (for AE31) or mass absorption cross-sections (for AE33) at 7 channels (370, 470, 520, 590, 660, 880, 950 nm). It should be noted that we adopted the acronym MAC throughout this paper to collectively denote both the mass attenuation cross-section and mass absorption cross-section throughout this study. The BC mass concentration reported by the Aethalometer is reported as equivalent black carbon (eBC) (Petzold et al., 2013; Savadkoohi et al., 2024). However, due to the
multiple scattering and loading effects, the reported eBC mass concentration is different from the ambient air. To correct the loading and multiple scattering effects, the parameterization scheme developed by Weingartner et al. (2003) was used for $b_{abs}$ reported by AE31. For AE33, the filter loading effect is compensated by a new real-time loading effect compensation algorithm and the multiple-scattering effect is corrected by a multiple-scattering enhancement correction ($C_0$) (Drinovec et al., 2015). The $C_0$ values are different between AE31 ($C_0$ = 2.14) and AE33 ($C_0$ = 1.39) due to the difference in tape materials (quartz fiber for
AE31 and tetrafluoroethylene-coated glass filter for AE33). Another factor that influences eBC mass concentration is the MAC, which is determined by the particle size, the mixing state of aerosol components, and the morphology of BC particles(Bond and Bergstrom, 2006; Bond et al., 2013; Yuan et al., 2021) and it shows spatial-temporal variations (Pandolfi et al., 2014; Zanatta et al., 2016; Grange et al., 2020; Guo et al., 2020; Savadkoohi et al., 2024). Therefore, using the nominal MAC from default settings (e.g., 16.6 m$^2$ g$^{-1}$ for AE31 and 7.77 m$^2$ g$^{-1}$ for AE 33 at 880 nm) without considering the MAC spatial-temporal
variability can result in misleading eBC estimations (Savadkoohi et al., 2024). To minimize the impact of MAC on the eBC mass concentration, the station-specific MAC values (**Table S1**) from previous studies (Zhang et al., 2008; Guo et al., 2020; Zheng et al., 2021), which utilized synchronous measurements of $b_{abs}$ and elemental carbon (EC) to derive MAC values.

Before data analysis, the raw data (5-min for AE31 and 1-min for AE33) acquired from the Aethalometers was checked and the outliers were removed according to the following procedures: (1) negative values of attenuation at time t and next
observation time (e.g., t + 5 min) were removed at all 7 wavelengths; (2) the observations which failed to comply with the





principle that the light absorption decreases with increasing of wavelength were removed and hourly mean values of remaining observations were calculated; (3) the outliers of the hourly time-series during the entire observation period were removed using the seasonal decomposition algorithm (Dancho and Vaughan, 2023). The absorption Ångström exponent of aerosol was calculated using the power law fitting between $b_{abs}$ and wavelengths at seven wavelengths and reported as $AAE_{370-950}$.

Observations with a *p*-value for fitting higher than 0.001 were considered as outliers and excluded from further analysis.

After data quality control, the data availability in each station of each year was calculated (**Fig. S2**). To get robust spatial-temporal variations of BC, the stations with data availability higher than 50% were used. To ensure data robustness, systematic screening criteria were applied: (1) long-term trend analysis required stations with more than 8 years of eBC observations, each year maintaining >50% data availability; (2) spatial-temporal pattern analysis included stations with more than 2 years

of valid observations during 2015 and 2017, where annual data completeness exceeded 50%. According to these rules , 34 and 25 stations were selected for spatial-temporal and long-term trend analysis, respectively (**Fig. 1**). It should be noted that data from other stations with data availability higher than 50% was also used (e.g., **Section 4.1** for validation of the BC model simulations).

## 2.3 Aethalometer model and eBC source apportionment

BC is formed via the combustion processes of carbonaceous matters and it is mainly from residential, power, industrial, and transport sectors (Bond et al., 2013; Li et al., 2017; McDuffie et al., 2020). To apportion the sources of BC, an Aethalometer model using aerosol absorption measurements at two wavelengths was developed by Sandradewi et al. (2008). This method has been widely used in eBC source apportionment (Harrison et al., 2012; Zheng et al., 2019, 2020, 2021; Savadkoohi et al., 2023) and the results of this method are generally consistent with other methods, e.g., receptor models (Favez et al., 2010;

Herich et al., 2011; Liu et al., 2014). In this study, eBC was apportioned into solid fuel combustion ($eBC_{sf}$) and liquid fossil fuel combustion ($eBC_{lf}$) rather than biomass burning ($eBC_{bb}$) and fossil fuel combustion ($eBC_{ff}$) reported by previous studies conducted in China (Zheng et al., 2020; Wang et al., 2023; Xu et al., 2024; Fan et al., 2025). The rationale for differentiating eBC into solid fuel (e.g., coal and biomass) combustion and liquid fossil fuel combustion (e.g., vehicle emissions) was based on the following reasons. (1) Bottom-up BC emission inventory showed that BC from coal, solid bio-fuel, and liquid fossil

combustion contributed 28.6%, 19.6%, and 28.0%, respectively to its total emissions (**Fig. S3a**). The discrepancy between fractions of solid fuel combustion emission and the attribution of eBC to $eBC_{sf}$ results (48.2% *vs* 48.2 ± 7.14%) was smaller than that between biomass burning emission and the attribution of eBC to $eBC_{bb}$ results (19.6% *vs* 48.2 ± 7.14%). (2) Based on experimental measurements of the AAE conducted on various aerosol types (**Fig. S3b**), the discrepancy between AAE values for aerosols from biomass burning and coal combustion (1.76 ± 0.40 *vs* 1.47 ± 0.13) was smaller than that between liquid

fossil fuel and solid fossil fuel combustion (1.00 ± 0.04 *vs* 1.47 ± 0.13). Given these two reasons, eBC was apportioned into $eBC_{sf}$ and $eBC_{lf}$ in this study.

The two key issues of using the Aethalometer model are the selection of the wavelengths and the AAE for aerosol from liquid fossil fuel combustion ($AAE_{lf}$) and solid fuel combustion ($AAE_{sf}$). According to Zotter et al. (2017), wavelength of 470 nm as a near-ultraviolet wavelength is recommended while the choice between 950 nm and 880 nm in the near-infrared



is less critical. Following previous studies (Sandradewi et al., 2008; Zotter et al., 2017), wavelengths of 470 nm and 950 nm were chosen in this study. The selection of $AAE_{lf}$ and $AAE_{sf}$, however, is more complicated. The AAE is used to describe the wavelength dependence of aerosol absorption. Although the aerosol AAE is impacted by particle size (Gyawali et al., 2009; Liu et al., 2018), chemical composition (Flowers et al., 2010; You et al., 2016), and morphology (Liu et al., 2008; Li et al., 2016), it can act as a proxy of aerosol composition (Yang et al., 2009; Russell et al., 2010). As summarized in previous studies

(Helin et al., 2021; Zheng et al., 2021), aerosol from liquid fossil fuel combustion has an AAE value of $\tilde{1}.0$, while aerosol from solid fuel combustion is characterized by a higher AAE value. The different methods were used to constrain the optimal AAE combinations and it showed spatial heterogeneity (Fuller et al., 2014; Becerril-Valle et al., 2017; Zotter et al., 2017; Helin et al., 2018; Mousavi et al., 2019; Zheng et al., 2021). Due to the lack of auxiliary measurements (e.g., radiocarbon, levoglucosan, and potassium) to constrain the station-specific AAE combination, the measured aerosol AAE frequency distribution (Tobler

et al., 2021; Rovira et al., 2025; Navarro-Barboza et al., 2025; Savadkoohi et al., 2025) was used to select the station-specific $AAE_{lf}$ and $AAE_{sf}$ values here. Specifically, the $AAE_{lf}$ and $AAE_{sf}$ were determined as the 1st and 99th percentiles of aerosol AAE, respectively. It should be noted that aerosol AAE was determined by power law fitting between light absorptions and wavelengths and the fittings with determinate coefficient ($r^2$) less than 0.99 were excluded (Savadkoohi et al., 2025). By this method, the station-specific AAE values for the Aethalometer model were determined (**Table S2**). Compared to the default

AAE values (i.e., $AAE_{lf} = 1.0$ and $AAE_{sf} = 2.0$), the Aethalometer model results with station-specific AAE values increased and decreased the fractions of $eBC_{sf}$ and $eBC_{lf}$ by 96.5 ± 72.3%, and 26.3 ± 16.9%, respectively (**Table S2**). The eBC source apportionment results using the station-specific AAE values were more consistent with the fractions of BC from different types of fuel combustion in China (**Fig. S3a**).

## 2.4 Weather normalization of eBC and its sources

To normalize the impacts of meteorological conditions on variations of eBC and its sources, a machine learning method (Grange et al., 2018) was used in this study. According to a previous study (Zheng et al., 2023b), the random forest (RF) model with hyper-parameters tuned in model training and temporal variables excluded in weather normalization can reduce the bias of long-term trends of air pollutants. We adopted these recommendations to normalize the impacts of weather on BC and its sources at each station. The entire observation was divided into two parts: 70% of the dataset was used to train the model

and the remaining 30% dataset was used to check the model performance. The variables used to train the model included the meteorological conditions, temporal variables, and transport indicator. Specifically, the temperature at 2 m (t2m), surface pressure (sp), wind components at 10 m (u10, v10), relative humidity (rh), accumulated precipitation along the trajectory (apt), boundary layer height (blh), surface downward solar radiation (ssr), and total cloud cover (tcc) were used here. The temporal variables included Unix time (number of seconds since 1 January 1970) acting as a linear trend term, Julian day (day of

the year) as the seasonal term and the day of the week. The transportation indicator was the cluster category of air masses reaching observational site calculated by the HYSPLIT (Stein et al., 2015). Three key hyper-parameters including the number of the tree (ntree), the number of variables that may split at each node (mtry), and the minimum size of the terminal nodes (min.node.size) were tuned by random search with 5-fold cross-validation after 100 times evaluation. The search space was



100–1000, 1–13, and 1–13 for ntree, mtry, and min.node.size, respectively. After tuning, the optimal parameter combination
(**Table S3**) with the lowest root mean square error (RMSE) was used to train the RF model. Despite the differences in statistics
between the RF model with parameters tuned and untuned were not significant (**Fig. S4**), the tuned model could increase the
correlation coefficient, index of agreement and reduce the RMSE. For instance, the Pearson correlation coefficient (*r*) between
observations and modeled eBC increased from $0.64 \pm 0.07$ (untuned) to $0.66 \pm 0.07$ (tuned). Therefore, the tuned parameters
in **Table S3** were used to train the RF model. After training, the weather normalization of the daily eBC and its sources was
achieved by repeatedly re-sample predictors and predicting with the trained RF model. For each prediction, the explainable
variables (meteorological conditions) in model training were randomly sampled from the entire dataset without replacement.
This predictive process was repeated 1000 times and the arithmetic mean values of these predictions were considered as the
emission-related concentrations (Zheng et al., 2023a, b).

## 2.5 Auxiliary dataset, data analysis and visualization

The auxiliary dataset used in this study included the meteorological conditions reanalysis, surface BC mass concentration
from the CTM simulations, and BC emission inventories. The meteorological conditions were from the fifth generation of
the European Centre for Medium-Range Weather (ECMWF) reanalysis for the global climate and weather (ERA5) (Hersbach
et al., 2023). Monthly mean values in surface BC mass concentrations between 2008 and 2020 were from the Modern-Era
Retrospective analysis for Research and Applications, Version 2 (MERRA-2) (Global Modeling And Assimilation Office
and Pawson, 2015) and Tracking Air Pollution in China (TAP) (Liu et al., 2022), respectively. Additionally, monthly mean
concentrations of BC from AeroChemMIP endorsed in Coupled Model Intercomparison Project 6 (CMIP6) were used in this
study (**Table S4**). We utilized the historical experiment from CMIP6, which incorporates time-varying emissions of greenhouse
gases, aerosols, and ozone, as well as volcanic and solar forcing, spanning the period from 2008 to 2014. This dataset provides a
comprehensive representation of the key drivers of climate change, allowing us to assess the impacts of both anthropogenic and
natural factors on global climate patterns during this period. Yearly BC emission inventories in China were from several data-
sets including Copernicus Atmosphere Monitoring Servic (CAMS (Granier et al., 2019)), Community Emissions Data System
(CEDS (Hoesly et al., 2018)), Emissions Database for Global Atmospheric Research (EDGAR, version: V4.3.2 (Crippa et al.,
2018), version: V5 (Crippa et al., 2019), version: V6 (Monforti Ferrario et al., 2021)), Hemispheric Transport of Air Pollution
(HTAP, version: V3 (Crippa et al., 2023)), Multi-resolution Emission Inventory model for Climate and air pollution research
(MEIC, version:1.4 (Geng et al., 2024)).

Data analysis was conducted using *R* language (R Core Team, 2024). The outlier detection was conducted by the "*anomalize*"
package (Dancho and Vaughan, 2023). The "*rmweather*" package (Grange et al., 2018) and "*mlr3*" (Lang et al., 2019) and its
extensions was used for weather normalization analysis. The model performance statistics were calculated using "*openair*"
(Carslaw and Ropkins, 2012). The figures in this study were generated by "ggplot2" (Wickham, 2016) and its extensions.
Other packages (e.g., "*lubridate*", "*plyr*") used in this study are available at the Comprehensive R Archive Network (CRAN,
https://cran.r-project.org, last accessed: 2025/06/03).



## 3 Results

### 3.1 Spatial-temporal variations

**Table S5** summarizes the statistics of eBC mass concentrations, source apportionment results, and $AAE_{370-950}$ values at 34
monitoring stations during the study period (2015–2017). eBC, $eBC_{lf}$, $eBC_{sf}$ mass concentrations and $AAE_{370-950}$ were aver-
aged as $2.05 \pm 2.85$ µg m$^{-3}$, $1.08 \pm 1.73$ µg m$^{-3}$, $0.97 \pm 1.52$ µg m$^{-3}$, and $1.33 \pm 0.29$, respectively. **Fig. 2** shows the spatial
distributions of eBC mass concentrations, and their source apportionment results between 2015 and 2017. As shown in **Fig.
2a**, BC from MERRA-2 showed higher levels in the east of the "Hu-Line" compared to those in the west of the line. As shown
in **Fig. 2b**, ground observations also showed higher levels in eastern China ($2.25 \pm 2.82$ µg m$^{-3}$) than those in western China
($1.51 \pm 2.86$ µg m$^{-3}$). Similarly, the spatial distribution of eBC observations showed a higher level in northern China ($2.19 \pm
3.45$ µg m$^{-3}$) than those in southern China ($1.91 \pm 2.05$ µg m$^{-3}$). The eBC concentrations in the different types of stations also
showed spatial heterogeneity with the highest values in urban ($2.67 \pm 3.41$ µg m$^{-3}$), which were significantly higher than that
in rural ($1.25 \pm 1.50$ µg m$^{-3}$, $p < 0.05$) and baseline ($1.07 \pm 1.15$ µg m$^{-3}$, $p < 0.01$) stations. The percentages of solid fuel
combustion to eBC ranged between $33.5 \pm 24.9\%$ and $65.2 \pm 24.8\%$ with the highest mean value in baseline stations ($49.9 \pm
230$ 10.9%). Regarding the spatial distribution (**Fig. 2c**), $eBC_{sf}$ in eastern China ($49.7 \pm 7.43\%$) and northern China ($50.0 \pm 7.25\%$)
were higher than those in western China ($44.4 \pm 4.55\%$) and southern China ($46.5 \pm 6.77\%$) without statistical differences. As
shown in **Fig. 2d**, the station-specific $AAE_{370-950}$ values were in the range of $1.11 \pm 0.13$–$1.66 \pm 0.40$ and the urban stations
had the lowest mean values ($1.29 \pm 0.11$) than those in baseline stations ($1.40 \pm 0.09$) and rural stations ($1.40 \pm 0.16$). Similar
to the spatial distribution of $eBC_{sf}$, $AAE_{370-950}$ in northern China ($1.43 \pm 0.12$) were significantly higher ($p < 0.001$) than those
in southern China ($1.24 \pm 0.07$) and $AAE_{370-950}$ in western China ($1.44 \pm 0.13$) were also significantly higher than those in
eastern China ($1.30 \pm 0.12$) at 0.01 levels.

The monthly variations of eBC, $eBC_{sf}$, and $AAE_{370-950}$ are shown in **Fig. 3**. Generally, the lowest values of eBC ($1.46 \pm
1.72$ µg m$^{-3}$), $eBC_{sf}$ ($36.3 \pm 24.7\%$), and $AAE_{370-950}$ ($1.21 \pm 0.28$) occurred during the summer (JJA) and the highest values
presented during winter (DJF). The higher values during winter were related to unfavorable meteorological conditions that
hindered the dispersion of surface BC. The higher demand for heating also contributed to the higher BC levels in winter, as
evidenced by higher percentages of $eBC_{sf}$ ($60.2 \pm 22.7\%$) and higher $AAE_{370-950}$ value ($1.45 \pm 0.26$). Some stations (e.g.,
58370 and 58448), however, showed the highest eBC mass concentrations during May and July. These anomalously high eBC
levels were related to biomass burning during the summer harvest season proved by high $eBC_{sf}$ percentages. Previous studies
also showed these stations or regions were impacted by biomass burning during summer harvest (Zha et al., 2014; Zhou et al.,
2019). Another station with high levels of $AAE_{370-950}$ during summer was Tazhong (51747), which was largely impacted by
dusty weather from March to July (Zhou et al., 2023).

### 3.2 Long-term trends from 2008 to 2020

**Figure 4** shows the time-series of eBC, $eBC_{lf}$, and $eBC_{sf}$ from 2008 to 2020. Despite eBC and its sources showing overall
decreasing trends from 2008 to 2020, the year with peak values differed between different types of stations. As shown in **Fig.**





**4a**, the annual mean concentrations of eBC and its sources at urban stations increased from 2008 to 2012 and then showed a downward trend to 2020. The inter-annual variations of eBC, eBC$_{lf}$, and eBC$_{sf}$ at rural stations showed peak values in 2009 and then continuously decreased until 2020. At baseline stations, eBC and its sources also showed downward trends with less variations compared to those at urban and rural stations. Using the Theil-Sen method, the slopes of eBC and its sources in each station were calculated. As listed in **Table S6**, most stations had decreasing trends with mean slope of $-0.17 \pm 0.20$ µg m$^{-3}$

yr$^{-1}$ for eBC, $-0.12 \pm 0.14$ µg m$^{-3}$ yr$^{-1}$ for eBC$_{lf}$, and $-0.06 \pm 0.09$ µg m$^{-3}$ yr$^{-1}$ for eBC$_{sf}$, respectively. Regarding the trends in the different types of stations (**Fig. 4b**), eBC in urban stations (N = 14) showed the fast-decreasing rate with a slope of $-0.20 \pm 0.23$ µg m$^{-3}$ yr$^{-1}$, which was quicker than that in rural stations ($-0.17 \pm 0.21$ µg m$^{-3}$ yr$^{-1}$, N = 6) and baseline stations ($-0.10 \pm 0.07$ µg m$^{-3}$ yr$^{-1}$, N = 5). Similarly, as shown in **Fig. 4d**, eBC$_{lf}$ also showed the fastest decreasing rates in urban stations ($-0.15 \pm 0.16$ µg m$^{-3}$ yr$^{-1}$), followed by rural stations ($-0.09 \pm 0.12$ µg m$^{-3}$ yr$^{-1}$) and baseline stations ($-0.05 \pm 0.04$

µg m$^{-3}$ yr$^{-1}$). The decreasing rates of eBC$_{sf}$ in baseline ($-0.05 \pm 0.04$ µg m$^{-3}$ yr$^{-1}$) and urban stations ($-0.06 \pm 0.10$ µg m$^{-3}$ yr$^{-1}$) were larger than those in rural stations ($-0.08 \pm 0.09$ µg m$^{-3}$ yr$^{-1}$). The differences in trends of eBC, eBC$_{lf}$, and eBC$_{sf}$ between southern/western China and northern/eastern China showed no statistical significance but with higher decreasing rates in southern China and western China (**Fig. S5**).

### 3.3 Comparison with other observations

To better understand the levels of eBC and its characteristics in China, the results in this study were compared to other regions with the same observational period (2015-2017) at the global scale (see **Text S1** for details). **Fig. 5** shows the global distributions of mass concentrations of eBC, eBC$_{sf}$ percentages, and AAE values from previous studies or observation networks during 2015 and 2017. Despite differences among the sampling inlets and monitoring instruments (**Table S7**), the mean values of eBC observations in China ($2.04 \pm 1.53$ µg m$^{-3}$, N = 35) were significantly lower than those in India ($6.30 \pm 4.12$ µg m$^{-3}$, N = 23) at

0.001 levels, but significantly higher than those in the UK ($1.22 \pm 1.10$ µg m$^{-3}$, N = 14, $p < 0.05$), Spain ($1.07 \pm 0.98$ µg m$^{-3}$, N = 9, $p < 0.05$), US ($0.83 \pm 0.63$ µg m$^{-3}$, N = 65, $p < 0.001$), Germany ($0.74 \pm 0.64$ µg m$^{-3}$, N = 13, $p < 0.001$), and Switzerland ($0.61 \pm 0.47$ µg m$^{-3}$, N = 5, $p < 0.001$). Regarding the BC from solid fuel combustion derived from the Aethalometer model, the results are influenced by the choice of wavelength pair and AAE combination. The other factors such as station types (e.g., background *vs* traffic), instruments, and sampling inlets also have impacts on the Aethalometer model results. The comparison

here used the AAE combination of 1.0 for AAE$_{lf}$, 2.0 for AAE$_{sf}$, and wavelength pair of 370 nm and 880 nm to minimize the influences of AAE combination and wavelength pair selection on source apportionment results. As showed in **Fig. 5e**, eBC$_{sf}$ had the highest contribution in the US ($30.2 \pm 12.8\%$, N = 25), which was higher than that in this study ($28.1 \pm 10.9\%$, N = 34). The lowest contribution of eBC$_{sf}$ was found in Spain ($13.3 \pm 5.51\%$, N = 3), which was lower than that in China ($p < 0.05$). Similar to the spatial distribution of eBC$_{sf}$, AAE also showed the highest value in the US ($1.40 \pm 13.7\%$, N = 26) and

the lowest value in Spain ($1.21 \pm 0.06$, N = 3).

The comparison of eBC trends between this study and other observations from 2008 to 2020 is shown in **Fig. 6**. Results from 88 stations (34 in this study and 54 from other observations) showed the widely decreasing trends of eBC in most stations (N = 83) with mean trends of $-4.36 \pm 3.06\%$ yr$^{-1}$. The slopes for eBC from 2008 to 2020 showed the fastest decreasing trend in



the UK ($-5.94 \pm 2.26\%$ yr$^{-1}$, N = 12) and the slowest decreasing trends in the US ($-2.61 \pm 2.04\%$ yr$^{-1}$, N = 18). The eBC
decreasing rates in Germany ($-4.93 \pm 2.51\%$ yr$^{-1}$, N= 16) and Switzerland ($-5.66 \pm 5.44\%$ yr$^{-1}$, N = 7) were also faster than
this study of China ($-3.98 \pm 3.41\%$ yr$^{-1}$, N = 25). Compared to the developed countries (UK, Germany, and Switzerland),
China showed higher eBC levels but low decreasing rates, suggesting the huge BC emission reduction potential in China.

## 4   Discussion

### 4.1   Comparison with CTM simulations

Due to the lack of long-term global-scale observations, knowledge of atmospheric BC abundance continues to rely on CTM
simulations. Therefore, it is important to evaluate the performance of CTMs in simulating the spatial-temporal variations of
BC. Owing to the availability of long-term BC observations in China, the BC simulation results from different models (see
**Section 2.5** for details) were evaluated in this study. Regarding model performance in simulating the spatial distribution of
BC, these models showed high values in East China, which were consistent with ground observations (**Fig. 7**). The $r$ between
mean values of simulations and observations from 2008 to 2014 ranged from 0.51 (MRI) to 0.70 (TAP). Compared to the
observations, the CTMs underestimated the surface BC mass concentrations by 80.8% (GISS) to 42.1% (UKEMS1) in China
from 2008 to 2014. The underestimation of surface BC concentrations by CTMs has been widely reported elsewhere (Koch
et al., 2009; Fu et al., 2012; Mao et al., 2016; Qi and Wang, 2019). Among the different types of stations, the CTMs showed
the highest performance in reproducing the surface BC concentrations at baseline stations ($r = 0.83 \pm 0.07$, slope = $0.88 \pm$
$0.42$), followed by urban ($r = 0.65 \pm 0.10$, slope = $0.44 \pm 0.26$), and rural stations ($r = 0.20 \pm 0.10$, slope = $0.13 \pm 0.08$). The
higher performance of CTMs in simulating BC in urban areas compared to rural areas is related to uncertainty in BC emissions.
For instance, BC is mainly from vehicle exhaust in urban areas, and is dominated by biomass burning and coal combustion in
rural regions. The uncertainty in BC emission factors for gasoline and diesel vehicles is lower than that for biomass burning
(Rönkkö et al., 2023). As a result, BC emissions in urban areas show less uncertainty (Zhao et al., 2011; Zhang et al., 2020),
and simulations from CTMs produce less deviation from observations in urban areas.

Regarding the temporal variations of BC in China, the simulations from CMIP6, MERRA-2, and TAP showed similar
monthly variations, with higher concentrations in winter and lower values in summer (**Fig. 8a**). The $r$ values between CTMs
and observations ranged from 0.63 (GISS) to 0.87 (MERRA-2) on a monthly resolution. On the annual scale (**Fig. 8b**), except
for TAP, annual concentrations of BC from CMIP6 historical experiments and MERRA-2 showed increasing trends from 2008
to 2014, while the ground observations and TAP showed decreasing trends in China. Except for TAP and MERRA-2, the
correlations between observations and CMIP6 results were negative at the annual scale (**Fig. 8d**), suggesting these models
failed to reproduce the yearly variations of surface BC. The upward trends of BC from CMIP6 historical experiments and
MERRA-2 were related to their inter-annual BC emissions, which also showed increasing trends from 2008 to 2014 in China
(**Fig. S6**). Similarly, the downward trends of surface BC concentrations from TAP were also in accordance with their emissions
(MEIC), which showed a decreasing trend from 2008 to 2014 (**Fig. S6**). The opposite BC trends between ground observations
and model simulations from CMIP6 in this study were similar to previous studies (Zhang et al., 2019b; Ramachandran et al.,



2022). For instance, aerosol optical depth from observations showed a decrease, while CMIP6 simulations showed increasing trends during 2002–2018 in China (Ramachandran et al., 2022). Therefore, caution should be exercised when interpreting the temporal variations of aerosols using simulations from CMIP6.

Due to the chemical inertness of BC particles, the deviations between observation and simulations arise from uncertainties in two aspects: emissions and deposition (Fan et al., 2018, 2022). The underestimation of BC emissions in China results in lower simulations of BC by CTMs. For instance, BC simulations were underestimated by 35%–60% in different seasons using the bottom-up BC emissions in China while the CTM showed better performance with BC underestimated only by 22%–9% using the top-down constrained emissions (1.59 times higher than bottom-up emissions) (Fu et al., 2012). Similarly, the RMSE for

annual BC simulation decreased by 31.6% if the BC emissions in China increased by 1.8 times (Wang et al., 2016). Using the MEIC instead of emissions inventory for IPCC AR5 experiments (BC emissions increased by 13.4%), the CTM improved the BC concentrations by 42.6% (Fan et al., 2018). Despite the increase of BC emissions in CTMs, the models still underestimated the BC concentrations, suggesting an overestimation of BC deposition within CTMs. For instance, a better agreement between observations and simulations was achieved by reducing the BC dry deposition velocity by 50% (Huang et al., 2010). Similarly,

using the lower dry deposition, the CTM simulations resulted in a lower global mean BC dry deposition flux and a higher correlation between simulations and observations (Wu et al., 2018). The observations also suggested an overestimation of BC aging degree in the CTMs, and sensitive simulations showed that slowing down the BC aging degree in model simulation can reduce the large model bias in simulation of surface BC concentration over China (Shen et al., 2023).

To identify which factor (uncertainty in emission or deposition) makes a higher contribution to the uncertainty of BC simu-

lation, we used the importance of each variable in RF model building (**Section 2.4**). As shown in **Fig. S7**, generally, date_unix was the variable with the greatest importance for the prediction of eBC, $eBC_{lf}$, and $eBC_{sf}$ in the RF model. Additionally, Julian day (seasonal term) had the third largest importance for the prediction of eBC and $eBC_{lf}$, while temperature showed higher importance for the prediction of $eBC_{sf}$. These explanatory variables can serve as proxies of emissions of BC, e.g., more biomass burning during colder months for heating. The explanatory variables related to deposition (e.g., wind speed, precipitation),

however, had less importance for prediction. Therefore, the emissions were more important than deposition when explaining BC concentrations.

### 4.2 Weather normalized concentrations: an indicator of BC emissions

Using the weather normalization method, the concentrations of eBC and its two sources ($eBC_{lf}$ and $eBC_{sf}$) were normalized. Compared to the eBC observations, a higher correlation between weather-normalized eBC concentrations and BC emissions

was found (**Fig. S8**), suggesting that the normalized eBC concentration can better reflect emission variations. Therefore, the weather-normalized eBC concentration was used here to discuss its relationship with BC emissions. As shown in **Fig. 9a**, except for MEIC, BC from the other emission inventories showed increasing trends since 2008, reaching peak values in certain years and then decreasing. In contrast, BC emissions from MEIC showed a continuous decrease since 2008, and their inter-annual variation was more consistent with weather-normalized eBC concentrations. In 2010, China began to fight against air

pollution, and the Action Plan on the Prevention and Control of Air Pollution and the Three-Year Action Plan were implemented





during 2013–2017 and 2018–2020, respectively. As a result of these actions, emission standards and activity levels changed over time. These factors are considered in the MEIC framework, which employs a technology-driven methodology to track China's energy statistics and technological advancements, reflecting shifts in emission characteristics across various sectors, fuels, products, combustion/process technologies, and emission control technologies in recent years (Li et al., 2017; Zheng

et al., 2018; Geng et al., 2024). Therefore, the BC emissions from MEIC coincided more closely with weather-normalized concentrations. The BC emissions from other inventories, such as CEDS, are not calibrated; instead, they are derived from default emissions using the Speciated Pollutant Emission Wizard, due to the limited availability of national BC inventory estimates in other countries. The similar reduction rates of surface BC concentrations and emissions (53.6% *vs* 51.9%) and their similar inter-annual variations from 2008 to 2020 in China also suggest that the normalized BC concentrations can be

used as a proxy to scale BC emissions in recent years. Updating bottom-up BC emissions is time-consuming, as it requires collecting activity data (e.g., energy consumption). As summarized in **Table S7** and mentioned above, there are several BC observation networks worldwide (e.g., UK, US, Germany, and India); the normalized BC observations can be used to calibrate the total BC emissions in these countries.

To further explore whether the weather-normalized BC source contributions can be used as an indicator of BC emissions

from different fuel types, the BC emissions from MEIC were used here. The sectors in MEIC (industrial, power, residential, and transportation) were further broken down into fossil fuel (liquid fuel + coal) and biomass burning according to (Wang et al., 2012). As shown in **Fig. 9b**, the variations of $eBC_{lf}$ concentrations from the Aethalometer model aligned with their emissions, showing a continuous reduction from 2008 to 2020. In contrast, the inter-annual variation of $eBC_{sf}$ was not in line with its emissions, with a small peak in emissions in 2013, while its concentration continuously decreased since 2008. The higher

correlation between $eBC_{lf}$ and its emissions compared to that between $eBC_{sf}$ and solid fuel emissions (**Fig. S9**) suggests higher uncertainty in BC emissions from solid fuel combustion, as discussed in **Section 4.1** and previous studies (Zhao et al., 2013; Zhang et al., 2020). Therefore, to further improve the accuracy of BC emission inventories, more attention should be paid to the solid fuel combustion sub-sector, e.g., focusing on surveys of activity rates and measurements of local emission factors (Li et al., 2017; Jiang et al., 2024).

## 375 4.3 Drivers of black carbon variations in China

Using the weather-normalized method, the contributions of emissions and meteorology to the inter-annual variations of air pollutants can be quantified (Chen et al., 2019; Zheng et al., 2020, 2023b). Results from **Fig. 10a** indicate that the reductions of eBC, $eBC_{lf}$, and $eBC_{sf}$ were dominated by emission-related reductions, with relative contributions of 89%, 87%, and 62%, respectively, implying the effectiveness of human efforts in BC emission reduction. The results here are in line with previous

studies showing that the inter-annual variations of air pollutants were dominated by emission variations (Chen et al., 2019; Cheng et al., 2019; Zhang et al., 2019a; Zheng et al., 2020, 2024). Compared to eBC and $eBC_{lf}$, $eBC_{sf}$ was more strongly impacted by variations in meteorological conditions, e.g., more coal and biomass were consumed for heating during colder months. To better understand which sources dominated the reduction of BC, the results from the Aethalometer model were used here. As shown in **Fig. 10b**, the slopes of the fittings between $eBC_{sf}$ and $eBC_{lf}$ in baseline and rural stations were higher





than 1, suggesting the dominant role of BC emission reduction from solid fuel in the decreases of ambient eBC concentrations in these two types of regions. In contrast, the slope between eBC$_{sf}$ and eBC$_{lf}$ in urban stations was lower than 1, suggesting the dominant role of BC emission reductions from liquid fuel (e.g., vehicle emissions) in the decrease of eBC concentration in urban areas. The control measures related to BC emission reductions include stricter vehicle emission/industrial standards, eliminating outdated industrial capacity, prohibiting agricultural residue burning and scattered coal combustion, replacing coal

with natural gas and electricity, etc. (Zheng et al., 2018). The focus of BC mitigation measures, however, differs between rural and urban areas. In urban areas, BC emission reductions are related to stricter vehicle emission standards, improved fuel quality, and eliminating outdated industrial capacity (Xu et al., 2018; Zheng et al., 2018; Wang et al., 2020; Zhang et al., 2021), which result in less BC emissions from liquid fuel. In rural areas, BC emission reductions result from prohibiting agricultural residue burning, household stove upgrading, and energy switching, e.g., from coal to natural gas and electricity (Zhu et al.,

2019; Meng et al., 2021; Shen et al., 2022). These measures lead to BC emission reductions from biomass burning and coal combustion. The results here are in line with the differences in BC emission reduction measures between city and rural areas, with fast decreases of eBC$_{sf}$ in rural areas and eBC$_{lf}$ in urban regions. Due to the limitation of the Aethalometer model, the BC sources are apportioned into eBC$_{sf}$ and eBC$_{lf}$, while the contributions of solid fossil fuel (e.g., coal) and biomass cannot be further separated from eBC$_{sf}$ (Zheng et al., 2021). To further understand which types of fuels (e.g., biomass, coal, liquid

fuel) dominated the BC reductions in China, we used the estimated BC emissions from different fuels derived from MEIC. As shown in Fig. 10**c**, BC from coal combustion decreased by 0.38 Tg from 2008 to 2020 in China, followed by liquid fuel (0.16 Tg), and biomass burning (0.09 Tg). Therefore, emission reduction from coal combustion dominated the BC reduction in China, with a contribution of 60.3%.

## 5  Summary and implications

In this study, 13 years of continuous measurements of black carbon aerosol were analyzed from 48 stations in China. Using station-specific AAE$_{lf}$ and AAE$_{sf}$, the sources of eBC were apportioned. The levels, spatial-temporal characteristics, and trends of black carbon aerosol were reported, and the key findings are listed as follows.

(1) Observations of black carbon aerosols from 2015–2017 showed averages of $2.05 \pm 2.85$ µg m$^{-3}$, $1.08 \pm 1.73$ µg m$^{-3}$, $0.97 \pm 1.52$ µg m$^{-3}$, and $1.33 \pm 0.29$ for eBC, eBC$_{lf}$, eBC$_{sf}$, and AAE$_{370-950}$, respectively, in China.

(2) Spatial distributions of eBC and eBC$_{sf}$ were higher in eastern and northern China compared to western and southern regions. Among different types of stations, urban stations exhibited the highest eBC values, while baseline stations had the highest fractions of eBC$_{sf}$ and AAE$_{370-950}$. The seasonal variations of eBC, eBC$_{sf}$, and AAE$_{370-950}$ typically showed the lowest values in summer and the highest in winter. However, some stations recorded abnormally high eBC levels during summer, which were associated with biomass burning and dusty weather events.

(3) Long-term trends of eBC, eBC$_{sf}$, and eBC$_{lf}$ indicated reductions from 2008 to 2020, with mean slopes of $-0.17 \pm 0.20$ µg m$^{-3}$ yr$^{-1}$ for eBC, $-0.12 \pm 0.14$ µg m$^{-3}$ yr$^{-1}$ for eBC$_{lf}$, and $-0.06 \pm 0.09$ µg m$^{-3}$ yr$^{-1}$ for eBC$_{sf}$ on a national scale. These trends exhibited spatial heterogeneity across station types. Urban stations experienced the fastest declines for eBC and eBC$_{lf}$,



while rural stations showed the highest rate of decline for $eBC_{sf}$. By normalizing concentrations to account for weather effects, the contributions of meteorological changes and emission reductions to the observed reductions in eBC concentrations were

quantified. The results indicated that emission reductions were the dominant factor in the decrease of eBC concentrations, contributing 89% for eBC, 87% for $eBC_{lf}$, and 62% for $eBC_{sf}$. In terms of anthropogenic drivers, the contributions to eBC reductions varied by station type: emission reductions from $eBC_{sf}$ were the primary driver at baseline and rural stations, while at urban stations, emission reductions from $eBC_{lf}$ had the greatest impact.

This study provides valuable insights from China's long-term observations of BC aerosol, with significant implications for

refining emission inventories and reducing model uncertainties, as detailed below.

(1) The comparison between eBC observations and simulations from various CTMs revealed that most models performed poorly due to inadequate estimations of inter-annual variations in BC emissions. In contrast, the TAP simulation showed strong performance in capturing both spatial and temporal variations of BC. Therefore, using BC emissions from MEIC is recommended for modeling BC concentrations in China. Although all models effectively simulated the spatial distribution of

BC, they generally underestimated BC levels, especially at rural stations. This underestimation suggests that enhancing the representation of BC emissions and reducing BC deposition rates could improve model accuracy.

(2) Compared to eBC observations, the inter-annual variations of weather-normalized eBC concentrations showed a stronger correlation with BC emissions, suggesting that these normalized BC observations could be used to refine the calibration of total BC emissions in recent years. Further analysis revealed a higher correlation between $eBC_{lf}$ and BC from liquid fossil

fuel emissions, while the correlation between $eBC_{sf}$ and BC from solid fuel combustion was weaker. To reduce uncertainties in BC emissions, greater attention should be given to the solid fuel combustion sub-sector, including the collection of real BC emission factors from biomass and coal burning, as well as updating activity data in rural areas.

*Data availability.* The authors do not have the right to share the ground BC observations dataset; The BC from TAP can be downloaded from http://tapdata.org.cn. BC mass concentrations from MERRA-2 can be downloaded from https://disc.gsfc.nasa.gov/datasets; Surface

BC observations in UK and USA can be accessed via https://uk-air.defra.gov.uk and https://www.epa.gov/outdoor-air-quality-data. The light absorption coefficient to calculate BC mass concentration and AAE values are from EBAS (https://ebas-data.nilu.no/Default.aspx); Historical simulations of surface BC during 2008–2014 from CMIP6 are available from https://esgf-node.llnl.gov/projects/cmip6/. The BC emission inventory used in this study can be downloaded from https://eccad.aeris-data.fr, http://inventory.pku.edu.cn, and http://meicmodel.org.cn. Surface meteorological parameters from ERA5 are available at the Climate Data Store (https://cds.climate.copernicus.eu/cdsapp#!/home).

GDAS1 to drive HYSPLIT is available at https://www.ready.noaa.gov/data/archives/gdas1/.

*Code and data availability.* The code and data used to produce all figures are available from the Huang Zheng (zhengh@wust.edu.cn) and corresponding authors under reasonable request.



*Author contributions.* HZ conducted the data analysis, data visualization, and wrote the paper. SK designed and edited the paper. DD provided the eBC observations in China and MS processed the pan-European eBC observations. CS, MZ, and RMH reviewed and commented on the paper. All co-authors reviewed and commented on the paper.

*Competing interests.* The authors declare that they have no known competing financial interests or personal relationships that could have appeared to influence the work reported in this paper

*Acknowledgements.* This study is financially supported by the National Natural Science Foundation of China (42307147, 42307151, 41830965), the Key Program of Ministry of Science and Technology of the People's Republic of China (grant nos. 2016YFA0602002). This study uses data that has received funding from the RI-URBANS project (Research Infrastructures Services Reinforcing Air Quality Monitoring Capacities in European Urban & Industrial Areas), European Union's Horizon 2020 research and innovation program, Green Deal, European Commission, contract 101036245. We greatly thank the staff from the Chinese Meteorological Administration for maintaining the instruments.



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



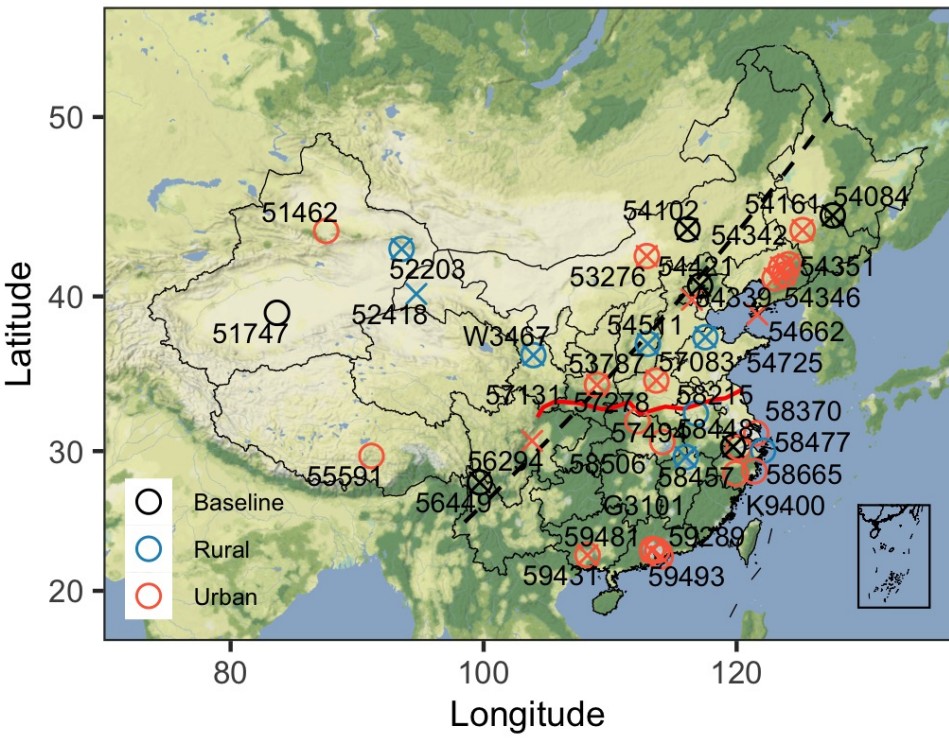

**Figure 1.** Spatial distribution of ground black carbon monitoring stations. The circles and crosses represent the stations used for spatial-temporal variation analysis between 2015 and 2017 (N = 34) and long-term analysis from 2008 to 2020 (N = 25), respectively. The colors of the dots indicate different types of stations. The black diagonal line represents the "Hu-Line," which separates China into eastern and western parts. The red polyline represents the "Qinling-Huaihe Line," which separates China into northern and southern parts.







**Figure 2.** Spatial distributions of mean values of BC concentrations, source apportionment results, and absorption Ångström exponent between 2015 and 2017 (**a**), and their box plots (**b–d**) showing station-specific values for different types of stations, including baseline (B), rural (R), and urban (U), as well as different regions in China: eastern China (EC), western China (WC), southern China (SC), and northern China (NC). The sizes and colors of the dots in panel **a** correspond to the mean values of AAE and eBC, respectively. NS, *, **, and *** in panels **b–d** indicate differences between two paired groups that are not significant ($p > 0.05$) or significant at the 0.05, 0.01, and 0.001 levels, respectively.





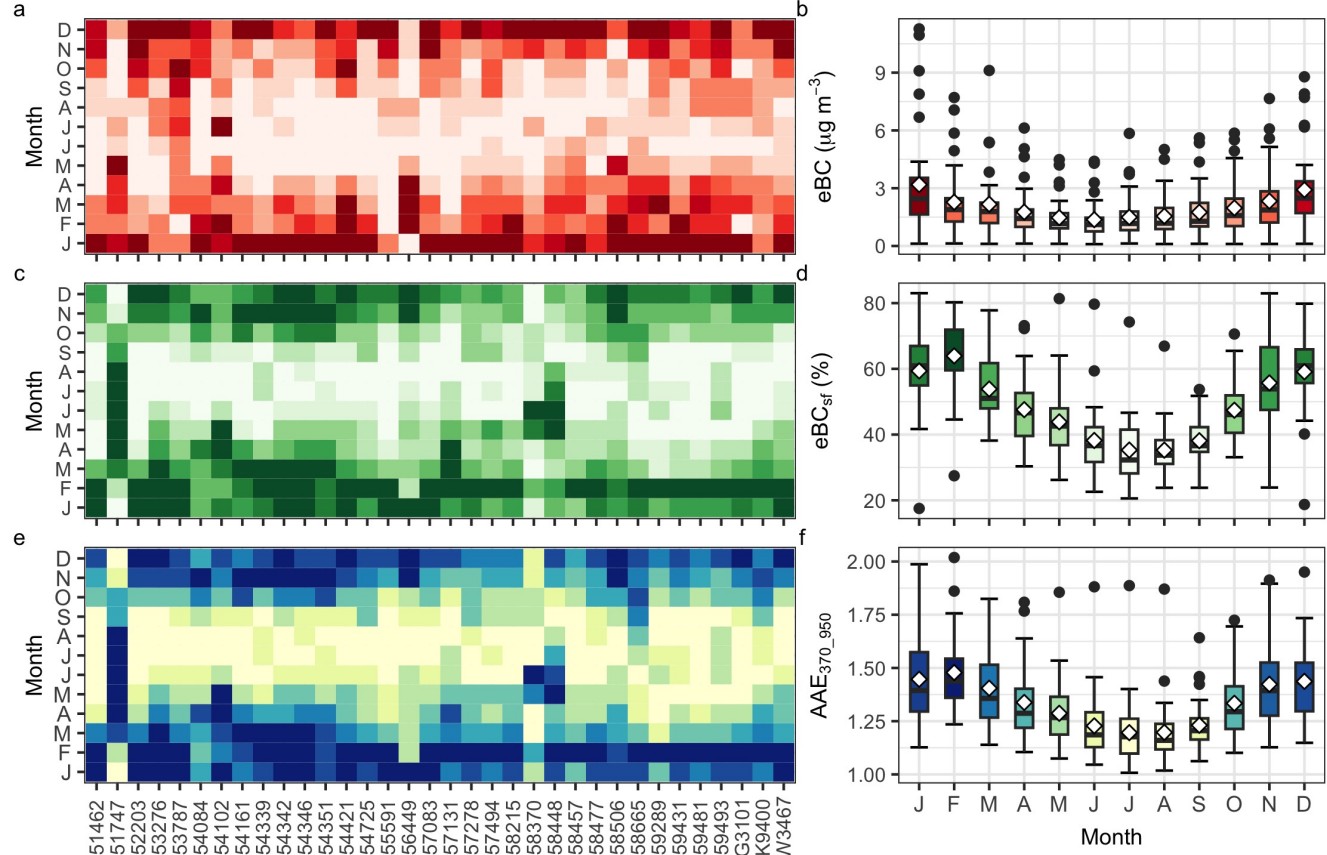

**Figure 3.** Monthly variations of black carbon concentrations (eBC, **a**, **b**), the fraction of BC from solid fuel combustion (eBC$_{sf}$, **c**, **d**), and the absorption Ångström exponent calculated by power law fitting at seven wavelengths (AAE$_{370-950}$, **e**, **f**). The filled grids in the left panels represent the scaled monthly values, ranging from 0 to 1 to better show the station-specific monthly variations. Note that the darker the color represents the higher the value.





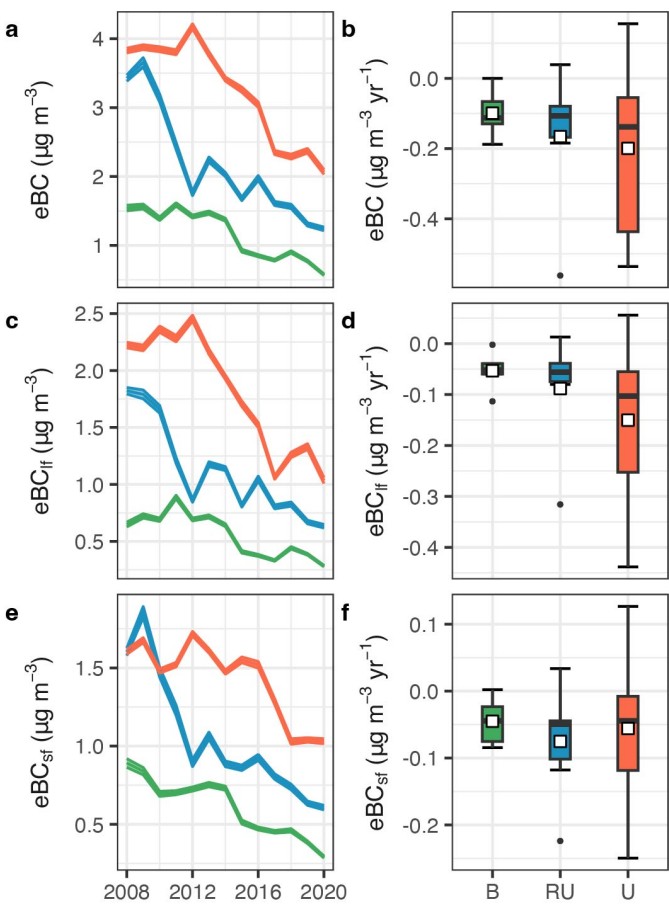

**Figure 4.** Annual concentrations of black carbon (**a**), black carbon from liquid fossil fuel combustion (eBC$_{lf}$, **c**), and solid fuel combustion (eBC$_{sf}$, **e**) from 2008 to 2020 and box plots (**b**, **d**, **f**) of their trends in different types of stations including baseline (B), rural (RU), and urban (U) stations. The solid line in panel **a** represents the annual mean values, and the filled ribbons represent the 99% confidence intervals of mean values.



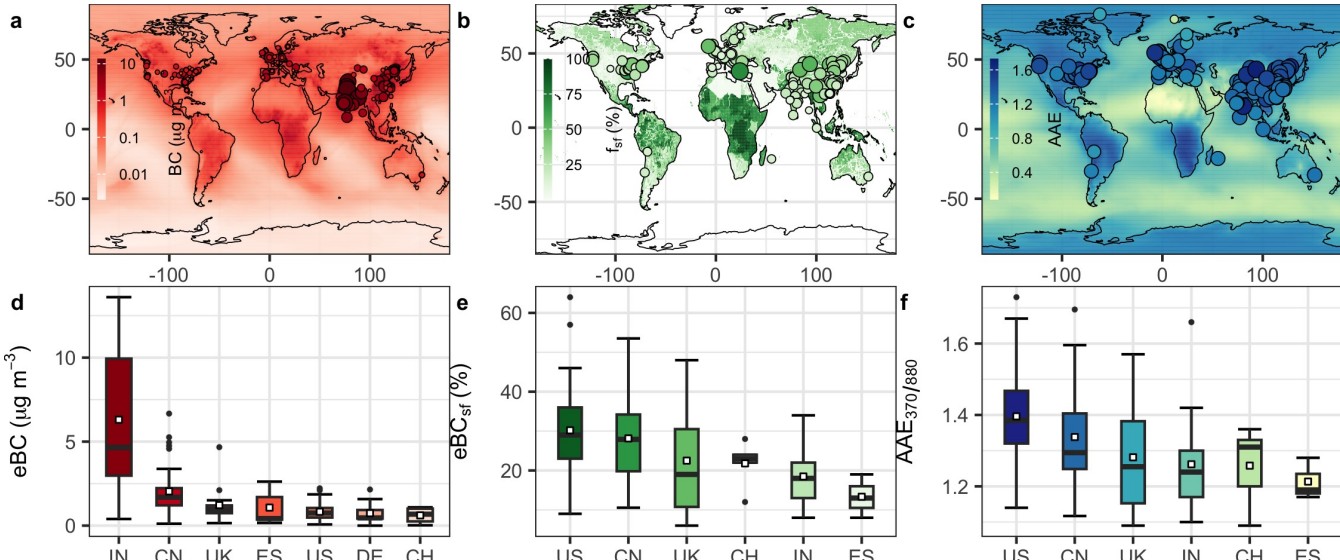

**Figure 5.** Spatial distributions and box plots in black carbon concentrations (**a**, **d**), black carbon from solid fuel combustion (**b**, **e**), and absorption Ångström exponent (**c**, **f**) during 2015–2017. The filled base maps of panels **a** and **c** are from MERRA-2 and the base map of panel **b** is from CEDS-MAPS (McDuffie et al., 2020). The dots in panels **a**–**c** represent the observation values (see **Table S7** for details) and the size of each dot is mapped to its value. Panels **d**–**f** are box plots of observations in different countries including India (IN), Switzerland (CH), China (CN), United Kingdom (UK), Spain (ES), Germany (DE), and the United States (US).





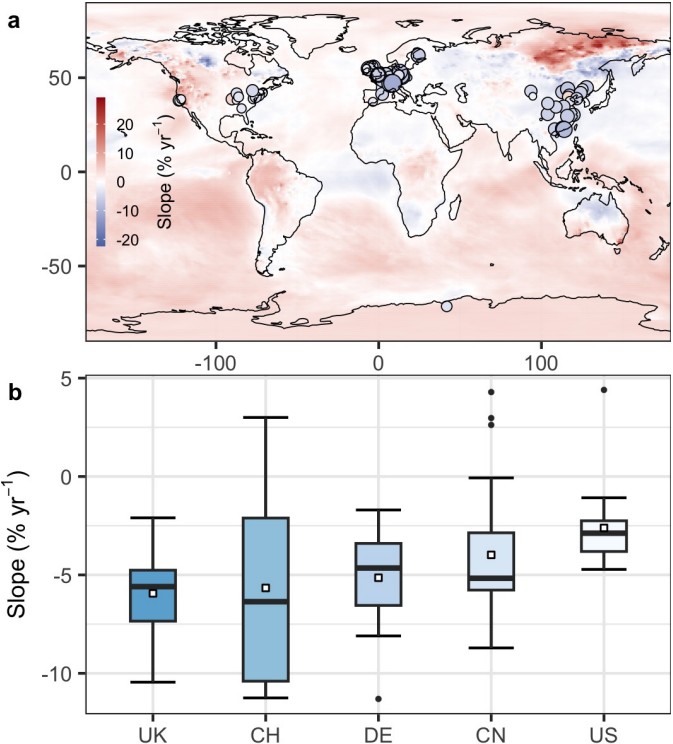

**Figure 6.** Spatial distribution of black carbon concentration trends (**a**) and box plots of trends in different countries (**b**) from 2008 to 2020. The size of dots in panel **a** corresponds to the absolute value of slope. The fill color in panel **b** corresponds to the mean slope value in each country. Detailed information about slopes at each station can be found in **Table S7**.



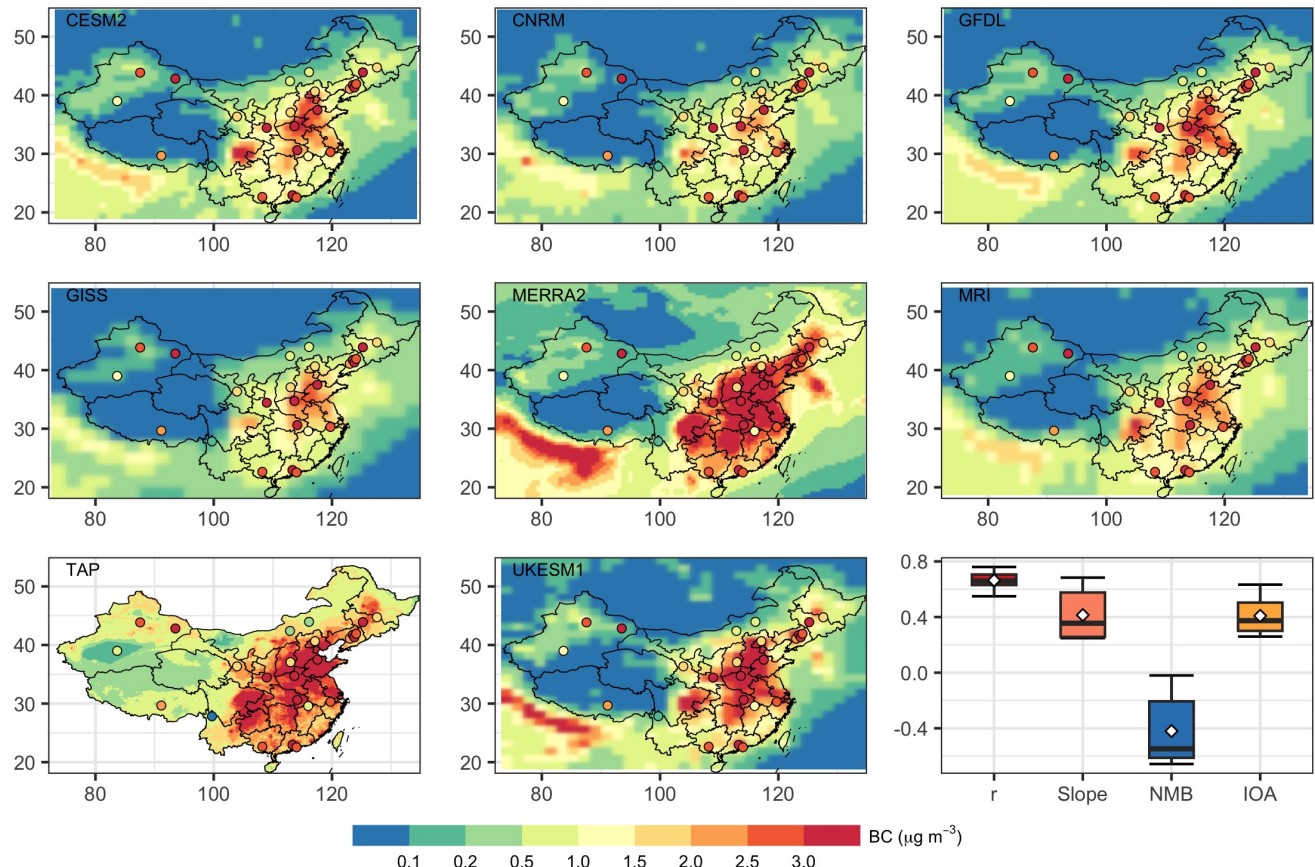

**Figure 7.** Spatial distributions of surface BC observations and simulations from different chemical transport models during 2008–2014 and the statistical metrics of model performance.





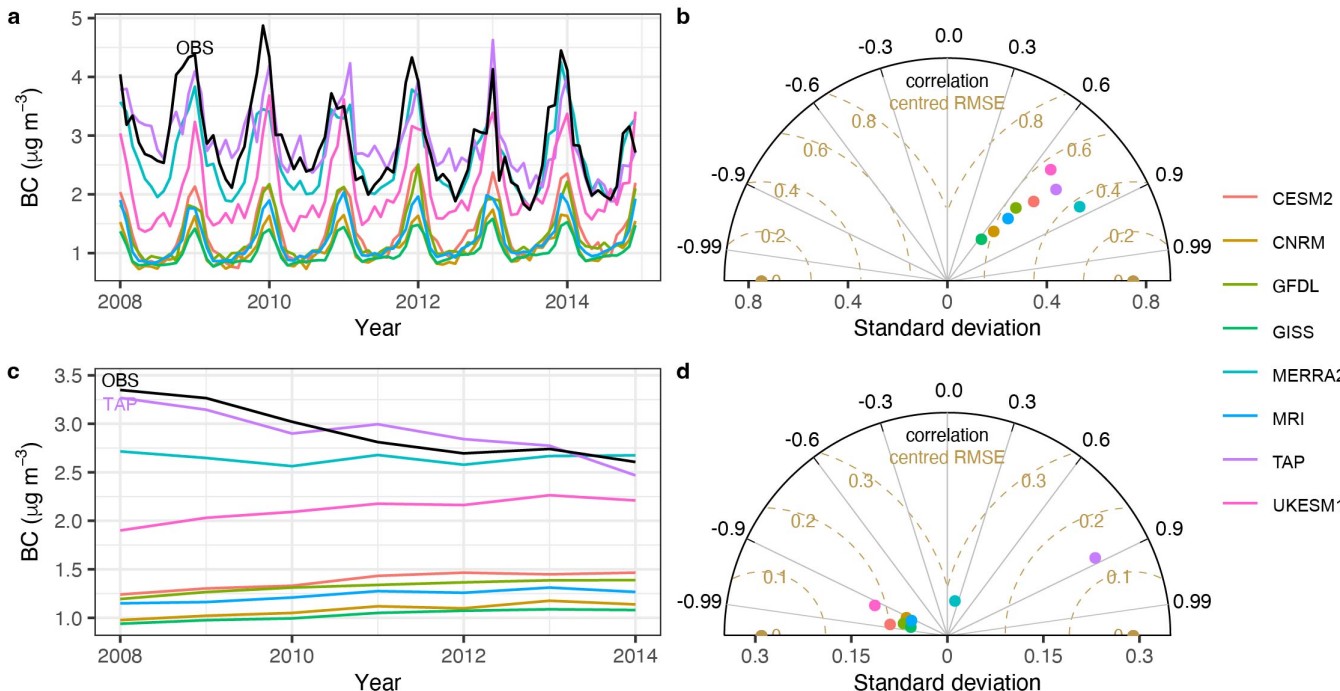

**Figure 8.** Monthly (**a**) and inter-annual (**c**) variations of BC observations and surface BC simulations from different chemical transport models during 2008–2014, and their Taylor Diagrams showing model performance at monthly (**b**) and yearly (**d**) scales.



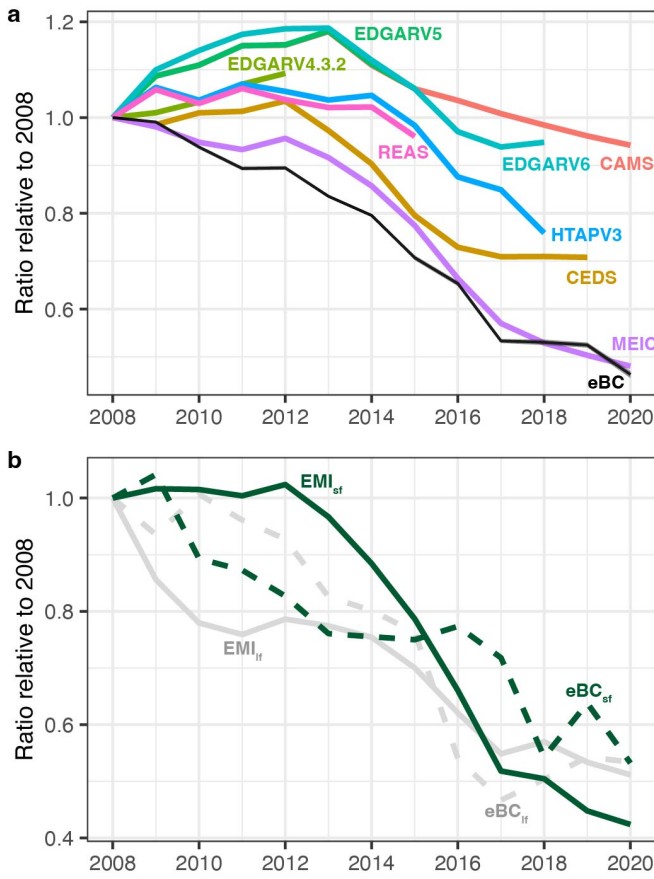

**Figure 9.** The inter-annual variations of black carbon emissions from different emission inventories and observations (eBC) during 2008–2020 (**a**) and the BC emissions from solid fuel combustion (EMI$_{sf}$) and liquid fossil fuel combustion (EMI$_{lf}$) derived from MEIC and the BC sources including solid fuel combustion (eBC$_{sf}$) and liquid fossil fuel combustion (eBC$_{lf}$) derived from the Aethalometer model (**b**).



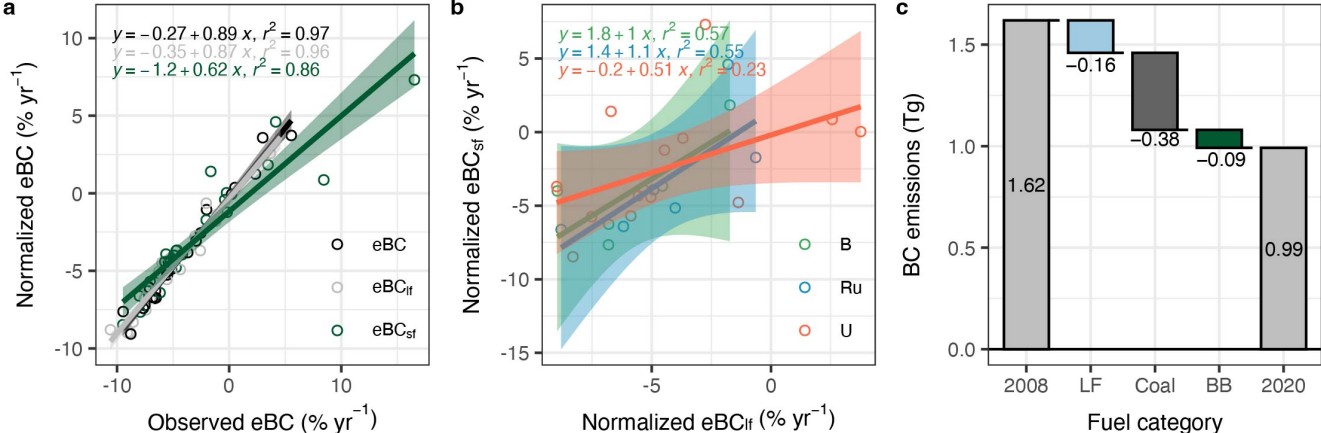

**Figure 10.** The drivers of BC reduction including contributions from emission variations (**a**), different sources (**b**), and fuel types including liquid fossil fuel (LF), coal combustion, and biomass burning (BB) (**c**) in China.