# Peer review of "Black carbon aerosols in China: Spatial-temporal variations and lessons from long-term atmospheric observations"

_EGUsphere, 2025_

## Author Comment (AC1)

Dear Editor and Referees:

Thanks for your comments towards improving our manuscript. We have addressed all your comments. Our author responses (AR) are marked in blue and the revisions in the manuscript are marked in red.

**Referee 1**

This study presents a comprehensive overview of Aethalometer based observations from CBNET, which includes measurements from 48 stations (including 23 urban, 18 rural sites and 7 background sites) across China during the period 2008-2020. The results are used to understand the discrepancies in chemical transport models and emission inventories. This study attempts to take a step further and address three problems: (1) What can we learn from the comparison between BC ground observations and CTM simulations; (2) Whether the inter-annual variations of BC can be used as an indicator of BC emissions; (3) Which factors dominated the variations of BC in China during the past 13 years. Thereby reducing the uncertainties in BC emissions and simulations from CTMs. The data analysis is robust and the different factors that need to be considered while handling aethalometer data were considered. The attempt to try out season specific AAE and de-seasonalizing the data using machine learning is novel.

AR: Thanks for your comments towards improving our manuscript and we have addressed the following issues you raised.

**Minor comments**

lines 113-120: The treatment of negative and following values: The manufacturer suggests retaining these values and doing a running mean before filtering the odd values. However, there is no general consensus on the methodology. Was a smoothing performed before or after removing the outliers? There would be a difference in the final values depending on the method followed. It would be good if this is also clearly stated.

AR: Thanks for your comments. We did not use the smoothing method to exclude the negative value outliers. For that method, it is difficult to apply an appropriate window size during the smoothing. Instead, the negative value at observation t and the following measurement (i.e., t + 5 min) were directly removed. If the negative value at time t was not removed, the observation at next time would be overestimated. As a response to your suggestion, we have added the following contents in the revised manuscript (Line 116-124 in the revised clean version and hereafter).

It should be noted that our method to process the negative value differed from the manufacturer's protocol, which recommends retaining negative values and applying smoothing techniques to exclude anomalous data points. In our method, the negative values were directly removed. This is because the Aethalometer employs the time-differential method to measure the optical attenuation (ATN) at the sampling point and subsequently converts ATN into BC mass concentration. The time-differential method implies that if the difference between the ATN in the current observation period (ATNt) and that of the previous observation period (e.g., ATNt-5 min) is negative, the calculated BC mass concentration for period t will be negative. Furthermore, the difference between the ATN in the next observation

period (e.g., ATNt+5 min) and that at time t (ATNt) will be overestimated, consequently resulting in an overestimation of the BC concentration at time t+5 min.

lines 121-128: While a criterion of minimum of 50% data points was adopted for passing quality control for long term analysis, it is not clear whether it is 50% points for a day/month/ annually. This needs to be made clear.

AR: Thanks for your comments and we have revised this part as follows (Line 127-130).

To get robust spatial- temporal variations of BC, the stations with annual data records exceeding 50% were used. The 50% threshold was selected to maximize the inclusion of monitoring stations while ensuring the collected data effectively captures the temporal variations of BC

Section 2.3: Station specific AAEs are determined as the 1st and 99th percentiles of the AAE distribution at the site. Although this avoids assuming a universal value for AAE it is still arguable that the selected values may or may not represent the actual conditions and fuel types. While the $eBC_{lf}$ (%) reduces up to 50% and $eBC_{sf}$ (%) almost triples at many stations compared to default values. These variations are huge and indicate that the $eBC_{lf}$ values are overestimated while the $eBC_{sf}$ underestimated by conventional methods. Are these values realistic or are the values blown up due to the use of % changes? I suggest including figures comparing the values with and without the application of this method in the supplementary for easy comparison and understanding.

AR: Thanks for your suggestion. Actually, we have compared the differences between the source apportionment results using the default and site-specific AAE combination in each site in Table S2 in the supplementary materials. Following your comment we have revised the difference into percentage calculated by default results (%) – SS results (%). We have also revised this part in the revised manuscript as follows (Line172-173).

Compared to the default AAE values ($AAE_{lf}$ = 1.0, $AAE_{sf}$ = 2.0), applying station-specific AAE values resulted in a mean change of 22.3 ± 8.10% in the fractions of $eBC_{sf}$ and $eBC_{lf}$ across most stations (**Table S2**).

Figures 1&2, Tables and other places

The station codes used in this study makes it too difficult to associate with individual stations. The observatories are denoted by long numbers making the figures clumsy and difficult to identify. I am curious to know if there a specific reason behind the station codes?

AR: Thanks for your comments and we have revised this figure as follows. Specifically, we have changed the station code into the acronyms that make it easier to be read. The full names are provided in Table S1.

[Figure]

Figure 1. Spatial distribution of ground-level black carbon monitoring stations. The dots with white circular frame and white crosses represent the stations used for spatial-temporal variation analysis between 2015 and 2017 (N = 34) and long-term analysis from 2008 to 2020 (N = 25), respectively. The colors of the dots indicate different types of stations. The black diagonal line represents the "Hu-Line," which separates China into eastern and western parts. The red polyline represents the "Qinling-Huaihe Line," which separates China into northern and southern parts. The full names of acronyms can be found in **Table S1**.

Figure 2: This figure is really impressive and rich in information content. On the downside we have long observatory names and randomly distributed pie charts. It is really difficult to find the pie charts corresponding to a particular location on the map. It would be way easier for the reader if they are arranged in ascending order as in Fig 3. Also, the figure caption does not say anything about the MERRA data used in the background. It is also not clear why the inset is used. Is it intended to only show the islands to the south of the mainland which got cut off?

AR: Thanks for your suggestion and we have revised this figure to make it clear. Specifically, the pie charts around the map are revised into donuts that are projected into each site in the map. Second, the codes have been revised as acronyms. Thirdly, we have added the information about the MERRA-2 date in the figure caption. The insert panel represents the South China Sea to make the territory of China complete.

[Figure]

**Figure 2** Spatial distributions of mean values of BC concentrations, sources of eBC including solid fuel (eBC_sf) and liquid fuel (eBC_lf) combustion, and absorption Ångström exponent between 2015 and 2017 (**a**), and their box plots (**b** − **d**) showing station-specific values for different types of stations, including baseline (B), rural (RU), and urban (U), as well as different regions in China: eastern China (EC), western China (WC), southern China (SC), and northern China (NC). The sizes and colors of the dots in panel (**a**) correspond to the mean values of AAE and eBC, respectively. The donuts in panel (**a**) represent the source apportionment results of eBC in each station. The data in the filled grid in panel (**a**) is from MERRA-2. NS, *, **, and *** in panels (**b**) – (**d**) indicate differences between two paired groups that are not significant ($p > 0.05$) or significant at the 0.05, 0.01, and 0.001 levels, respectively.

Figure 4: It would be good to add a legend here. As I go through this section, I am curious to know if any individual location showed a positive trend.

AR: Thanks for your comments and we have added the legend into this figure as shown below. The trend of each site is provided in Table S6, and some stations showed positive trends. We have extended the description in the revised manuscript as follow (Line 284-285).

As listed in **Table S6**, most stations showed decreasing trends, while only a few sites (e.g., Yushe in Shanxi province) exhibited an increasing trend.

[Figure]

Revised Fig. 4 Annual concentrations of black carbon (**a**), black carbon from liquid fossil fuel combustion ($eBC_{lf}$, **c**), and solid fuel combustion ($eBC_{sf}$, **e**) from 2008 to 2020 and box plots (**b, d, f**) of their trends in different types of stations including baseline (B), rural (RU), and urban (U) stations. The solid line in panel (**a**) represents the annual mean values, and the filled ribbons represent the 99% confidence intervals of mean values.

Section 3.3: If the eBC values for China are estimated using the SS AAE direct comparison with values from existing studies does not make much sense. The older studies use the conventional methods to estimate eBC while here the corrections pull down the eBC values.

AR: Thank you for your comments. The eBC source apportionment results are sensitive to the wavelength pair selection and AAE parameterization. Since the raw Aethalometer data (e.g., AE31 time-resolved signals) were unavailable, we could not correct for the loading and shading effects, which were known to bias AAE calculations. Consequently, we applied the default AAE combination ($AAE_{lf} = 1.0$, $AAE_{sf} = 2.0$) uniformly across all datasets, including our measurements and harmonized data from external networks/literature. Additionally, to maintain consistency with datasets limited to 370/880 nm absorption channels (e.g., US EPA monitoring), Section 3.3 analyses exclusively use this wavelength pair.

Figure and table captions.

The figures are not self-explanatory as key information is missing in the caption. It makes it difficult for the reader to easily find out key information.

AR: Thanks for your comments and we have revised these two figures by adding the legend and axis labels as shown below.

[Figure]

Revised Fig.5 Spatial distributions and box plots in black carbon concentrations (**a, d**), black carbon from solid fuel combustion (**b, e**), and absorption Ångström exponent (**c, f**}) during 2015–2017. The filled base maps of panels (**a**) and (**c**) are from MERRA-2 and the base map of panel (**b**) is from CEDS-MAPS (Mcduffie et al., 2020). The dots in panels (**a–c**}) represent the observation values (see **Table S7** for details) and the size of each dot is mapped to its value. Panels (**d–f**}) are box plots of observations in different countries including India (IN), Switzerland (CH), China (CN), United Kingdom (UK), Spain (ES), Germany (DE), and the United States (US).

[Figure]

Revised Fig. 6 Spatial distribution of black carbon concentration trends (**a**) and box plots of trends in different countries (**b**) from 2008 to 2020. The size of dots in panel (**a**) corresponds to the absolute value of slope. The fill color in panel (**b**) corresponds to the mean slope value in each country. Detailed information about slopes at each station can be found in **Table S7**.

Spelling errors

Table S3 Hyperparameter tunning

AR: Thanks, and we have corrected it.

Concluding comments: This study has made a great effort in comparing observations with CTMs. The authors find that models and outdated inventories fail represent BC over China leading to underestimations and increasing trends. While the station specific AAE is interesting, the method remains inherently oversimplified, as it does not fully capture complexities such as the distinction between coal and biomass sources, seasonal variability in BC lifetime, and mixing state effects that influence optical properties. While the percentile-based approach is practical, the arbitrariness of AAE selection cannot be entirely ruled out. A more robust validation such as through chemical tracer analysis (e.g., OC/EC or $^{14}$C) would be ideal, although such efforts may be beyond the scope of this study and could be considered for future work. I also feel that the random forest modeling performed in this study was a bit under utilised in the end. Pictorial representation of the changes observed after using this model would make a good impact on the reader.

AR: Thanks for your comments. Regarding the validation of using the percentile-based station-specific AAE values to apportion the BC, you suggest the use of radiocarbon isotope to constrain source apportionment results. The other referee also mentioned this issue and as a response to your comments, we have extended the discussion of the reliability of source apportionment here by comparing our results with previous studies using the dual carbon isotopes as follows (Line 174-180).

Validation against dual-carbon isotope data from previous studies (Fang et al., 2018; Ni et al., 2018) supports the use of station-specific AAE values. As shown in **Fig. S4a**, compared to results using default AAE values, the $eBC_{sf}$ fraction calculated with station-specific AAE values showed a stronger Pearson correlation (r = 0.37 vs. 0.20) with the solid-fuel combustion fraction (coal + biomass) determined by dual-carbon isotopes. Seasonal patterns of $eBC_{sf}$ derived from the Aethalometer model were consis-tent with those from carbon isotope analysis (Fig. **S4b**). Therefore, station-specific AAE values were adopted for BC source apportionment in this study.

[Figure]

**Figure S4** Comparison of BC source apportionment between the Aethalometer model and dual carbon isotopes from previous studies (Fang et al., 2018; Ni et al., 2018). Panel (a) is a scatterplot between the fraction of BC from solid fuel ($f_{sf}$) derived from dual carbon isotopes and the Aethalometer model. Panel (b) is the seasonal variations of $f_{sf}$ derived from isotopes (gray bars) and Aethalometer model with default AAE (blue dots) and station-specific AAE (red dots).

Regarding the representation of the changes observed after using the RF model, we have added a figure to show the difference between observational and weather-normalized concentrations of BC as follows (Fig. R1) and it has been added into the revised supplementary materials as Fig. S12.

[Figure]

Fig. R1 Yearly variations of weather-normalized (EMI) and observational (OBS) concentrations of black carbon (a) and their changes (expressed as Δ) between 2020 and 2008 (b) at different type of stations.

We also extended the description of this figure in the revised manuscript as follows (Line 443-448).

As shown in **Fig. S12**, compared to the observations, the emission-related (weather-normalized) concentrations of BC were higher in each year. For instance, the national mean concentrations of emission-related and observed eBC in 2008 were 3.59 μg m$^{-3}$ and 3.47 μg m$^{-3}$, respectively, yielding meteorology-related concentrations of −0.12 μg m$^{-3}$. The negative value of the meteorology-related concentration indicates favorable meteorological conditions for reducing BC concentrations. Similarly, meteorology also contributed to lowering the concentrations of eBC$_{lf}$ and eBC$_{sf}$ at different types of stations (**Fig. S12**).

**Review of "Black carbon aerosols in China: Spatial-temporal variations and lessons from long-term atmospheric observations" by Zheng et al.** This study presents the characteristics of black carbon concentration variations in China using observational data from 2008 to 2020. The authors well describe the process of calculating black carbon concentrations through aethalometer measurements and explain the long-term variation characteristics of black carbon concentrations by comparing them with CTM and reanalysis data. Numerous papers have already been published regarding the long-term characteristics of black carbon concentrations in China, and many of these studies are cited in this paper. All studies consistently report decreases in black carbon concentrations due to air quality reduction policies. Therefore, it would be beneficial to clearly explain how the scientific findings (conclusions) newly reported in this study are distinctive compared to previous works.

AR: Thanks for your comments and suggestions. We have revised the conclusion to show the novelty of this study.

Additionally, while this study reports measurement results up to 2020, several recent papers have explained black carbon concentration variation characteristics through various factors including emissions and meteorological conditions, albeit regionally within China (e.g., https://www.nature.com/articles/s44407-025-00010-z). From this perspective, I suggest extending the analysis period of this study to include recent years beyond 2020.

AR: Thanks for your suggestion. The paper you listed focused on a city in Beijing, which was relatively easy to get the long-term observations till now. We can only get the observational data from 2008 to 2020 in whole China while the dataset beyond 2020 is not available to us. Sorry for it.

The study presents results of black carbon (eBC) concentrations observed through filter-based optical methods. The process of calculating eBC concentrations is clearly and well explained through citations of previous studies. However, for the methodology of dividing contributions into solid fuel combustion (eBCsf) and liquid fossil fuel combustion (eBClf), which represents a major finding of this study, detailed methodological presentation and sufficient validation of results are required. In particular, clear presentation of theoretical limitations and associated uncertainties in calculations based on AAE from multi-wavelength optical measurements is necessary.

AR: Thanks for your suggestion. We have extended the description of the uncertainty of the Aethalometer model as follows (Line 181-187).

While using station-specific AAE values derived from percentiles improved agreement with dual-carbon isotope results, the Aethalometer model still has several uncertainties. These include sensitivity to wavelength selection, the choice between fixed and variable AAE values, measurement noise, interference from non-BC absorbing components,

source-specific variability, and limitations in temporal resolution (Zheng et al., 2021; Savadkoohi et al., 2025). The percentile-based approach is especially sensitive to sources with high brown carbon content (such as coal combustion) and complex source mixtures (Savadkoohi et al., 2025). Despite these limitations, this method provides a practical solution for eBC source apportionment in situations where ancillary data (e.g., radiocarbon) are not available.

Furthermore, many studies have explained the emission sources of black carbon or elemental carbon through carbon isotope analysis, determining contributions from solid fuel and liquid fossil fuel combustion. I believe the most critical aspect for publication of this study would be to validate the calculation results by comparing them with such previous research findings and demonstrating consistent results. For example, I suggest presenting comparisons between this study and carbon isotope analysis-based contribution assessments for specific locations and time periods in the main text.

AR: Thanks for your suggestion. We have searched the elemental carbon source apportionments published previously using the dual carbon isotope method and compared it with our Aethalometer model results. It should be noted that to match the study location and duration between previous studies and this study, many carbon isotope constrained results were discarded. Finally, comparison of three sites from two published papers was conducted. In the revised manuscript and we have extended the comparison as follows (Line 174-180).

Validation against dual-carbon isotope data from previous studies (Fang et al., 2018; Ni et al., 2018) supports the use of station-specific AAE values. As shown in **Fig. S4a**, compared to results using default AAE values, the $eBC_{sf}$ fraction calculated with station-specific AAE values showed a stronger Pearson correlation (r = 0.37 vs. 0.20) with the solid-fuel combustion fraction (coal + biomass) determined by dual-carbon isotopes. Seasonal patterns of $eBC_{sf}$ derived from the Aethalometer model were consis-tent with those from carbon isotope analysis (Fig. **S4b**). Therefore, station-specific AAE values were adopted for BC source apportionment in this study.

[Figure]

**Figure S4** Comparison of BC source apportionment between the Aethalometer model and dual carbon isotopes from previous studies (Fang et al., 2018; Ni et al., 2018). Panel (a) is a scatterplot between the fraction of BC from solid fuel ($f_{sf}$) derived from dual carbon isotopes and the Aethalometer model. Panel (b) is the seasonal variations of $f_{sf}$ derived from isotopes (gray bars) and Aethalometer model with default AAE (blue dots) and station-specific AAE (red dots).

Meanwhile, Sections 3.1 and 3.2 present concentrations, contributions, seasonal changes, and annual variation characteristics in a rather straightforward manner compared to the observational and analytical results, lacking detailed explanations of why such contributions and spatiotemporal variations occur. Such explanations would be necessary to establish some differentiation from existing studies.

AR: Thanks for your comments and we have extended this part as follows (Line 265-276).

To investigate the drivers of spatial-temporal variations in BC, we performed a correlation analysis between BC concentrations, emissions, and meteorological parameters. BC emissions from different fuel types were obtained from CEDS-MAPS, which divided sources into solid bio-fuel, coal, liquid fuel plus natural gas, and process emissions (McDuffie et al., 2020). Meteorological data for 2015–2017 were sourced from ERA5 reanalysis dataset (Hersbach et al., 2023). As shown in **Fig. S6**, mean BC concentrations during 2015–2017 were positively correlated with emissions at most stations (N = 32), with Pearson coefficients of 0.65 for eBC, 0.56 for $eBC_{sf}$, and 0.58 for $eBC_{lf.}$ This significant correlation indicates that the spatial distribution of BC closely follows its emission patterns, which are higher in northern and eastern China (Wang et al., 2012). On a monthly scale, BC concentrations were positively correlated with emissions and negatively correlated with temperature, boundary layer height, and wind speed at most stations (**Fig. S6**). Elevated BC levels in colder months are attributed to increased heating-related emissions and less favorable dispersion conditions (lower boundary layer height and wind speed) (Zhang et al., 2019b; Xie et al., 2025). Overall, both emission patterns and meteorological factors jointly control the spatial-temporal variability of BC in China.

In Section 3.3, the authors discuss results by simply presenting comparisons with observation points in other countries. For such international data, more detailed descriptions of the observational data used are needed, and the justification for why such comparisons are necessary in this study should be clearly presented. Given the purpose of explaining black carbon concentration changes in China, it appears that this entire section could be removed without detriment to the study.

AR: Thanks for your comments. The detailed information upon the observations from other networks is provided in the **Text S1**. We have also explained the value of such a comparison between this study and other international observations as follows (Line 324-331).

This international comparison provides a global context for evaluating China's BC concentrations and trends, allowing for direct comparison with both developing countries (e.g., India) and developed countries (e.g., the United States, United King- 325 dom, Germany, Switzerland). The results highlight that while China achieved reductions in BC concentration, its concentrations still remain higher, and its rates of decline were slower than those observed in developed nations. In these countries, rapid BC reductions have been achieved through the implementation of stringent regulations (Luoma et al., 2021) and the adoption of cleaner technologies (Font and Fuller, 2016). These findings underscore the need for continued policy efforts in China, par- ticularly targeting industrial and residential sources, and support the refinement of emission inventories to further reduce BC emissions.

The content of Section 4.1 has also been mentioned in many previous studies. The most important conclusion the authors wish to convey in this section is not clearly evident. For example, clear explanation is required regarding whether the contribution analysis results based on aethalometer in this study possess the accuracy required for CTM simulation validation and improvement, or whether the authors intend to discuss methods for improving CTM simulation accuracy.

AR: Thanks for your comments and we have extended the method to improve CTM simulation accuracy as follows (Line 387-397).

The comparison between the observed BC concentration values and simulations suggested that to improve the model's accuracy, more attention should be given to updating the BC emission inventory. Here, we only discuss how to use BC obser- vation data for updating the BC emission inventory, rather than attempting to complete this task. In practice, the BC emission inventory, observed concentrations, FLEXPART simulations, and Bayesian inversion framework can be integrated to update BC emissions (Evangeliou et al., 2018, 2021; Jia et al., 2021). For instance, Evangeliou et al. (2021) used BC observations, outdated prior BC emissions, and an inversion framework to estimate BC emissions during the COVID-19 lockdown in Europe. Their results showed a 23% decline in BC emissions during the lockdown compared to the same period in the previous five years (2015–2019). Similarly, Jia et al. (2021) estimated a maximum weekly emission reduction of 70% in eastern China and 48% in northern China from January 1 to March 10, 2020. These studies showed improved accuracy of model simulations using posterior BC emissions

compared to the prior emission inventory, highlighting the potential to reduce BC simulation uncertainty by incorporating BC observations and prior emissions.

The conclusions in Section 4.2 "Weather normalized concentrations: an indicator of BC emissions" also appear ambiguous. More detailed explanation is required regarding whether this explains the previous observational results or whether emission source accuracy improvement is needed from a modeling perspective.

AR: Thanks for your suggestion and we have extended the discussion as follows (Line 431-440).

Accuracy in sectoral emissions is important to improve the performance of CTMs. Previous studies have demonstrated that uncertainties in BC emission inventory, originating from incomplete activity data, unrepresentative emission factors, and sectoral inconsistencies (Li et al., 2017), will propagate significant biases into chemical transport models. For instance, sectoral uncertainties are pronounced for residential and industrial sources in developing countries due to inconsistent emission factors and activity data. From a modeling perspective, these uncertainties dominate over other processes (e.g., aging, deposition) in near-source regions, while transport and removal uncertainties amplify errors in remote areas (Vignati et al., 2010). Another study showed that BC simulation was overestimated by a factor of 2.22, which can be attributed to the highest domestic emissions during winter in East Asia (Ikeda et al., 2022). Results from a top-down study showed the largest bias in residential and transportation sectors during January and July, and the bias of BC was reduced using the posterior emission inventory compared to the prior emission inventory (Zhao et al., 2019).

The causes of black carbon concentration reduction ultimately mentioned in Section 4.3 are all results that have been discussed in previous studies. Specific description of what this study presents differentially compared to previous research is needed.

AR: Thanks for your comments, and we have extended the discussion as follows (Line 476-482).

In line with previous long-term observations of BC conducted in urban areas (Chen et al., 2016; Zheng et al., 2020; Sun et al., 2022; Xie et al., 2025; Abulimiti et al., 2025), these studies identified the predominant driving factor for BC reduction was the reduced emissions from fossil fuel combustion (e.g., coal and petroleum). Representative examples include studies in Wuhan (Zheng et al., 2020), Beijing (Xie et al., 2025), and Nanjing (Abulimiti et al., 2025). Unlike previous studies, this research also analyzed the driving factors behind BC reduction at rural and background sites. The results demonstrate that the decrease in BC concentrations is mainly attributed to reduced

emissions from both coal and biomass combustion in rural and background areas. These findings represent novel conclusions not previously reported in earlier research.

---

## Author Response (AR2)

**Dear Editor and Referees:**

Thanks for your comments towards improving our manuscript. We have addressed all your comments. Our author responses (AR) are marked in blue and the revisions in the manuscript are marked in red. It should be noted the comments raised by previous referee are marked in *italic*.

Consider the following comments from one of the reviewers:

Overall, the paper has been thoroughly revised in response to the comments; however, several comments appear to require further clarification. The main concerns regarding the revised manuscript are as follows:

AR: Thanks for your comments and we have addressed all your comments as follows.

1. Regarding the previous comment about the relationship between studies based on long-term observational data from the Beijing site and this study dealing with data observed across many regions in China, the essence of the question pertains to the differences in major conclusions between this study and such previous research. The point was not merely to indicate that long-term observational data from a single site, like Beijing, is easily accessible. Additional clarification on this matter appears necessary.

AR: Thanks for your comments and we have extended this part by comparing our results with most recent long-term observations (Line 331-337 in the revised manuscript and hereafter).

Long-term observational studies across China have widely reported decreases in BC concentrations over various periods. For example, BC concentrations in Beijing declined at an average rate of 0.19 μg m-3 per year between 2013 and 2022 (Xie et al., 2025). An annual decrease of 0.10 μg m-3 was observed in Nanjing from 2014 to 2021 (Abulimiti et al., 2025), while a trend of –0.12 μg m-3 per year was recorded at Mt. Lushan from 2008 to 2022 (Liu et al., 2025). These single-site observations consistently highlight the effectiveness of air pollution control policies in reducing BC concentrations. Our study reinforces these findings from a broader spatial perspective, confirming that the observed reduction in BC levels can be attributed to the implementation of national regulatory measures, such as the Clean Air Action Plan and the Three-Year Action Plan.

2. A clear answer has not been provided regarding the previous question about the contributions of solid fuel combustion (eBCsf) and liquid fossil fuel combustion (eBClf). Sufficient responses and their incorporation into the paper are essential.

AR: Thanks for your comments and we have evaluated the accuracy of BC source apportionment results, and the details can be found in AR3 below.

3. Additional review is needed regarding the author's response to the following previously asked question:

Furthermore, many studies have explained the emission sources of black carbon or elemental carbon through carbon isotope analysis, determining contributions from solid fuel and liquid fossil fuel combustion. I believe the most critical

aspect for publication of this study would be to validate the calculation results by comparing them with such previous research findings and demonstrating consistent results. For example, I suggest presenting comparisons between this study and carbon isotope analysis-based contribution assessments for specific locations and time periods in the main text.

Examining the author's response (Fig. S4), the aethalometer-based analysis results of this study show very large differences from dual carbon isotopes analysis results depending on time and location. The differences are too substantial to be explained simply by correlation coefficients (r). Therefore, the distinction between solid fuel combustion (eBCsf) and liquid fossil fuel combustion (eBClf) based on aethalometer data in this study remains uncertain, and this reviewer finds it difficult to fully trust the results of this study. More detailed explanations are absolutely necessary regarding how the characteristics of aerosols from solid fuel and liquid fossil fuel combustion differ, and how and why wavelength-dependent differences in aethalometer observational data appear.

AR: Thanks for your comments and we have extended the explanation related to your concerns as follows (Line 156-167).

The Aethalometer model operates on the assumption that light-absorbing aerosols originate predominantly from two source types (Sandradewi et al., 2008), e.g., liquid fuel combustion and solid fuel combustion, which are characterized by distinct Absorption Ångström Exponent (AAE) values. Aerosols derived from solid fuel combustion (AAEsf) generally exhibit higher and more variable AAE values compared to those from liquid fuel combustion (AAElf). The elevated AAEsf values are attributed to the presence of significant amounts of organic carbon species, which absorb strongly in the ultraviolet and lower visible wavelength ranges (Sandradewi et al., 2008). In contrast, the lower AAElf values result from the dominant contribution BC, which absorbs light broadly across the visible spectrum. These source-specific AAE characteristics are consistent with laboratory combustion experiments, which also report higher AAE values for solid fuel emissions and lower values for liquid fuel emissions (Olson et al., 2015). Consequently, measured ambient AAE values can serve as an indicator of dominant BC sources, e.g., lower AAE values are typically associated with environments strongly influenced by liquid fuel combustion, such as road tunnels (Blanco-Alegre et al., 2020).

In addition to the above question, clear discussion of the accuracy of the methodology for distinguishing contributions of solid fuel combustion (eBCsf) and liquid fossil fuel combustion (eBClf) is crucial for deriving the main results of this study, and therefore, the related content must be sufficiently explained and validated.

AR: Thanks for your comments and we have used the error propagation to discuss the accuracy of the BC source

**apportionment results as follows (Line 213-223).**

Due to the unknown of "real" BC source apportionment results, the accuracy of the Aethalometer model was assessed through error propagation (Martinsson et al., 2017; Zheng et al., 2020). The primary sources of uncertainty considered were the absorption coefficient and the AAE values. The uncertainty of  $b_{abs}$  was reported as 5% (Hansen, 2005) and this value was adopted for this study. For the uncertainty of AAE values, their uncertainties were defined as the difference between the values derived from the percentile method and the isotope-constrained values from Zotter et al. (2017), e.g., 0.90 for AAElf and 1.68 for AAEwb. This approach was justified as the mean optimal AAElf and AAEsf values obtained here  $(0.90 \pm 0.05 \text{ and } 1.70 \pm 0.23$ , respectively) were closely aligned with the literature values. The error propagation analysis revealed station-specific relative uncertainties in the range of 19.8%–53.8% for eBCsf and 23.7%–54.0% for eBClf (**Table S3**). The reported uncertainties here were comparable to previous studies, e.g., 41% for fossil fuel combustion and 42% for wood burning estimated by Martinsson et al. (2017). Despite the relatively high estimated uncertainty, the spatial and temporal patterns of BC sources resolved from the Aethalometer model were believed reliable here.

Table S3 Relative uncertainty of the Aethalometer model in each station using the propagation of errors

| Code | $eBC_{\rm sf} \\$ | $eBC_{lf}$ | Code | $\mathrm{eBC}_{\mathrm{sf}}$ | $eBC_{lf}$ |
|------|-------------------|------------|------|------------------------------|------------|
| UR   | 32.0%             | 35.3%      | XA   | 30.7%                        | 24.4%      |
| TZ   | 53.8%             | 25.6%      | XF   | 30.8%                        | 27.5%      |
| HM   | 41.8%             | 47.2%      | WH   | 30.4%                        | 35.6%      |
| ZR   | 32.6%             | 33.0%      | SX   | 37.7%                        | 34.3%      |
| YS   | 25.0%             | 27.4%      | PD   | 35.2%                        | 27.2%      |
| LFS  | 19.8%             | 40.8%      | LA   | 27.6%                        | 54.0%      |
| XL   | 29.0%             | 23.7%      | HAZ  | 53.8%                        | 33.8%      |
| CC   | 21.6%             | 51.8%      | DH   | 37.3%                        | 34.7%      |
| AS   | 24.3%             | 30.0%      | LUS  | 23.9%                        | 25.4%      |
| SY   | 24.8%             | 36.5%      | НЈ   | 39.8%                        | 29.6%      |
| BX   | 25.3%             | 47.4%      | DG   | 35.4%                        | 27.3%      |
| FS   | 22.7%             | 37.0%      | NN   | 30.4%                        | 45.8%      |
| SDZ  | 20.7%             | 29.4%      | PY   | 44.3%                        | 45.4%      |
| HUM  | 21.6%             | 32.6%      | SZ   | 49.1%                        | 48.6%      |
| LAS  | 24.2%             | 27.8%      | HZ   | 52.7%                        | 31.1%      |
| XGL  | 28.6%             | 23.9%      | LS   | 33.2%                        | 39.9%      |
| ZZ   | 33.2%             | 33.5%      | GL   | 30.5%                        | 30.8%      |

5. One of the original findings claimed by the authors is the factor contributing to black carbon reduction in rural and background areas of China. As the authors argue, this is clearly distinctive from previous studies that primarily reported black carbon concentration variation characteristics in urban areas. However, overall, the main results of this study largely align with results already reported in previous studies, and the authors explain their research results in various places through citations of such previous research. It is required that this study minimize discussion of similar results compared to previous studies and provide sufficient and detailed explanations of the distinctive factors of this study. For example, regarding the conclusion of the following sentence, it is necessary to provide other observational data or emissions demonstrating reduced coal and biomass use in rural and baseline areas: The results demonstrate that the decrease in BC concentrations is mainly attributed to reduced emissions from both coal and biomass combustion in rural and baseline areas. These findings represent novel conclusions not previously reported in earlier research. (L480-482).

AR: Thanks for your comments and we have provided the emission date to support the novelty of this study, and we have also extended the discussions as follows (Line 525-529).

Evidence from BC emission data further support these findings. As shown in **Fig. S13**, BC emissions derived from the MEIC inventory (Geng et al., 2024) for biomass burning, solid fuel combustion, and liquid fuel combustion exhibited widespread reductions across most regions of eastern China between 2008 and 2020. Temporally, emissions from these sources showed consistent decreasing trends from 2008 to 2020 at rural, baseline, and urban stations. For example, BC emissions from rural residential energy consumption in northern China decreased from 0.85 Tg in 2010 to 0.55 Tg in 2020 (Zhang et al., 2023).

**Figure S13** Changes in emission reduction of biomass burning (a), solid fuel combustion (b), and liquid fuel combustion (c) between 2008 and 2020 and their time series of BC emission amounts from different fuels compared to 2008 for different station types (d-f).

6. Additionally, it would be helpful to provide a clear definition of how "baseline area" is defined in the above sentence.

AR: Thanks for your comments. We have provided the description about the baseline station as follows (Line 98-101).

Baseline stations are established in areas remote from strong emission sources and human activities to monitor the long-range transport of BC aerosols and their natural emissions. For instance, the Lin'an (LA) baseline station is situated at an altitude of 139 m a.s.l. in Zhejiang province, positioned approximately 150 km northeast of Shanghai and 50 km west of Hangzhou city.

7. A response to this notification from the journal is required: Figures 1, 2, 7 and S1 may contain a territory that is disputed according to the United Nations. If and when the manuscript is accepted for final revised publication, you will be asked to choose one of the following options: (a) you could remove the disputed territory from the map and submit new figure files, or (b) we could add a statement that some figures contain disputed territories.

AR: Thanks for your notification and we choose **option b** that you could add a Disclaimer.

---

## Author Response (AR3)

**Dear Editor**

Thanks for your comments towards improving our manuscript. We have addressed all your comments. Our author responses (AR) are marked in blue and the revisions in the manuscript are marked in red.

Please follow the guidelines for abstract and the conclusion section: https://www.atmospheric-chemistry-and-physics.net/policies/guidelines for authors.html.

AR: Thanks for your suggestion and we have revised the abstract and conclusion section. Regarding the abstract, we have followed the instruction and added the research background and gap into the abstract. Additionally, the revised abstract is less than 250 words according to the instruction. Additionally, we have removed the repeated part in last version and the revised abstract (238 words) is listed as follows.

Black carbon (BC) significantly influences climate, air quality, and public health, and long-term observations are essential for understanding its adverse effects. While previous studies have primarily focused on spatiotemporal variations, deeper insights from such datasets remain uncovered. Using 13 years (2008-2020) of continuous measurements of equivalent black carbon (eBC) in China, this study reported the spatial-temporal variations of eBC and its sources, including solid fuel (eBCsf) and liquid fuel combustion (eBClf). The results showed that eBC and its sources exhibited higher concentrations in eastern and northern China compared to western and southern China. Seasonal variations of eBC and eBCsf generally showed lower values during summer and higher values during winter at most stations. Long-term trends indicated that eBC and eBClf decreased most rapidly at urban stations, while eBCsf declined faster at rural stations. Comparisons of eBC concentrations and trends between this study and global observations revealed higher eBC levels but lower reduction rates in China. These long-term observations showed that the model simulations performed well in simulating spatial distribution but poorly in capturing inter-annual variations. The weather-normalized eBC concentrations showed potential for adjusting emission estimates. The normalized results also suggested that emission control was the dominant driver of the BC reduction. This decrease was primarily driven by reductions from solid fuel combustion at rural and background stations. This study provides insights for reducing uncertainties in black carbon emission inventories and improving model performance in simulating surface concentrations.

Regarding the conclusion section, we have revised the Conclusion to summarize the main findings, including key quantitative results, and to discuss their implications for reducing uncertainty in future BC modeling. The structure has been adjusted to align with journal guidelines by consolidating the results and discussion into distinct sections. Minor content edits were also made for clarity and focus. The revised Conclusion is listed as follows.

In this study, 13 years of continuous measurements of black carbon aerosol were analyzed from 48 stations in China. Using station-specific AAEIf and AAEsf, the sources of eBC were apportioned. The levels, spatial-temporal

characteristics, and trends of black carbon aerosol were reported, and the key findings of this unique dataset are listed as follows.

Observations of black carbon aerosols from 2015–2017 showed averages of  $2.05 \pm 2.85 \,\mu g \, m^{-3}$ ,  $1.08 \pm 1.73 \,\mu g \, m^{-3}$ ,  $0.97 \pm 1.52 \,\mu g \, m^{-3}$ , and  $1.33 \pm 0.29$  for eBC, eBC1f, eBCsf, and AAE370-950, respectively, in China. Long-term trends of eBC, eBCsf, and eBClf indicated reductions from 2008 to 2020, with mean slopes of  $-0.17 \pm 0.20 \,\mu g \, m^{-3} \, yr^{-1}$  for eBC,  $-0.12 \pm 0.14 \,\mu g \, m^{-3} \, yr^{-1}$  for eBC1f, and  $-0.06 \pm 0.09 \,\mu g \, m^{-3} \, yr^{-1}$  for eBCsf on a national scale. Spatial distributions of eBC and eBCsf were higher in eastern and northern China compared to western and southern regions. Among different types of stations, urban stations exhibited the highest eBC values, while baseline stations had the highest fractions of eBCsf and AAE370-950. The seasonal variations of eBC, eBCsf, and AAE370-950 typically showed the lowest values in summer and the highest in winter. However, some stations recorded abnormally high eBC levels during summer, which were associated with biomass burning and dusty weather events. The spatial-temporal patterns of BC in China can be explained by its emission variations and meteorological conditions. The weather normalized BC concentrations showed that emission reductions were the primary driver, accounting for 89% of the decrease in eBCsf trends exceeding 1, indicating that reductions in liquid fuel combustion emissions were the main contributor to BC decreases in urban areas. In contrast, rural and baseline stations exhibited the fastest declines in eBCsf suggesting that emission reductions from solid fuel combustion were the dominant factor in these regions.

This study provides valuable insights from China's long-term observations of BC aerosol, with significant implications for refining emission inventories and reducing model uncertainties. The comparison between observations and simulations revealed that most models failed to capture the inter-annual variation while the simulation with MEIC inventory was capable to reproducing the spatial and temporal variations of BC. Therefore, BC emissions from MEIC is recommended for modeling BC concentrations in China. Furthermore, a systematic model underestimation of BC was found particularly at rural stations, which emphasized the need to refine both emission source strengths and deposition processes to improve model accuracy. This study showed that the weather normalization is a promising technique for refining BC emission inventories, as the weather normalized BC concentrations showed a stronger correlation with the reported BC emissions than raw observations. Further analysis revealed a higher correlation between emissions and concentrations of BC from liquid fuel combustion. To reduce uncertainties in BC emissions, a target effort to improving accuracy of emission from the solid fuel combustion subsector is needed, including the collection of accurate BC emission factors from biomass and coal burning, as well as updating activity data in rural areas.

**Notification to the authors:**

Please ensure that the colour schemes used in your maps and charts allow readers with colour vision deficiencies to correctly interpret your findings. Please check your figures using the Coblis – Color Blindness Simulator (https://www.color-blindness.com/coblis-color-blindness-simulator/) and revise the colour schemes accordingly with the next file upload request. -> Fig. 8, 9, 10

AR: Thanks for your reminder and we have revised the color schemes of Fig. 8, 9, and 10 as follows.

Revised Fig. 8.

Revised Fig.10.

---

## Author Response (AR4)

**Dear Editor,**

Thanks for your comments towards improving our manuscript. We have addressed all your comments. Our author responses (AR) are marked in blue and the revisions in the manuscript are marked in red.

A few technical corrections are needed:

Thanks, and we have revised the technical corrections as follows.

Notification to the authors from review file validation

1. Please ensure that the colour schemes used in your maps and charts allow readers with colour vision deficiencies to correctly interpret your findings. Please check your figures using the Coblis – Color Blindness Simulator (https://www.color-blindness.com/coblis-color-blindness-simulator/) and revise the colour schemes accordingly with the next file upload request. -> Fig. 10

AR: Thanks, and we have checked the color schemes of our figures using the Coblis. We have also revised the color schemes of Fig. 10 and using different shapes in panel a and b to help interpreting it.

2. Please note, if you used scientific abbreviations without giving the written-out explanation, these must be written out with the next file upload request.

AR: Thanks, and we have carefully checked the manuscript, and the full names of missing scientific abbreviations have been given as listed below.

Line 38-41 in the revised manuscript: Based on BC observations from 2010 to 2016 at three representative background sites in East Asia, Choi et al. (2020a) estimated the average 40 transport efficiency of BC to be 0.73, which was lower than the mean rate of 0.91 from the FLEXible PARTicle Lagrangian transport model (FLEXPART, version 10.4).

Line: 221-222 in the revised manuscript: The transport indicator was the cluster category of air masses reaching the observational site, calculated by the Hybrid Single-Particle Lagrangian Integrated Trajectory (HYSPLIT) (Stein et al., 2015)

Line 244-247 in the revised manuscript: Model simulations from the Community Earth System Model 2 (CESM2), Centre National de Recherches Météorologiques (CNRM), Geophysical Fluid Dynamics Laboratory (GFDL), NASA Goddard Institute for Space Studies (GISS), Japan Meteorological Research Institute (MRI), UK Earth System Model (UKESM1) were used here (see **Table S5** for details).

Figure 7 caption: The NMB and IOA represent normalized mean bias and index of agreement, respectively.

3. However, do not forget that there is a limit to characters (not words!) for "Short summary": it must be < 500 characters.

AR: Thanks, and we have checked the "Short summary", and the total characters is 471 (including blank).

Please change the title of the summary section to 'Conclusion'

AR: We have revised it.